# Predicting power outages caused by extratropical storms

Roope Tervo[1,*], Ilona Láng[1,*], Alexander Jung[2], and Antti Mäkelä[1]

[1]Finnish Meteorological Institute, B.O. 503, 00101 Helsinki, Finland
[2]Aalto University, Dept of Computer Science, B.O. 11000, 00076 Aalto, Finland
[*]These authors contributed equally to this work.

**Correspondence:** Roope Tervo (roope.tervo@fmi.fi)

**Abstract.** Strong winds induced by extratropical storms cause a large number of power outages, especially in highly forested countries such as Finland. Thus, predicting the impact of the storms is one of the key challenges for power grid operators. This article introduces a novel method to predict the storm severity for the power grid employing ERA5 reanalysis data combined with forest inventory. We start by identifying storm objects from wind gust and pressure fields by using contour lines of $15 \mathrm{~m~s}^{-1}$ and 1000 hPa respectively. The storm objects are then tracked and characterized with features derived from surface weather parameters and forest vegetation information. Finally, objects are classified with a supervised machine learning method based on how much damage to the power grid they are expected to cause. Random Forest Classifier, Support Vector Classifier, Naive Bayes, Gaussian Processes, and Multilayer Perceptron were evaluated for the classification task, Support Vector Classifier providing the best results.

*Copyright statement.* TEXT

## 1 Introduction

Strong winds caused by extratropical storms are among the most significant natural hazards in Europe, causing massive damage to the forests and society (e.g. Schelhaas et al. (2003); Schelhaas (2008); Ulbrich et al. (2008); Seidl et al. (2014); Valta et al. (2019)); extratropical storms are responsible for 53 percent of all losses related to natural hazards in Europe (Kron W., Schuck A., 2013). Such storms pose a huge challenge for power distribution companies in highly-forested countries such as Finland (Gardiner et al., 2010) where falling trees cause power outages for hundreds of thousands of customers every year (Niemelä, 2018). The windstorms create a significant risk for the power supply in Finland, which has over 90 000 kilometers of overhead lines (70 percent of it medium-voltage, 1-35 kV, network) passing through forest (Kufeoglu and Lehtonen, 2015). Between the years 2010 and 2018, on average 46 percent of all transmission faults in Finland were caused by extratropical storms (Finnish Energy, 2010-2018). During the years of the most damaging storms, 2011 and 2013, the share of windstorm damages of all fault causes was up to 69 percent (Finnish Energy, 2011, 2013). The need for managing power interruptions is even more urgent since the power suppliers in Finland are obliged to financially compensate customers of urban areas after 6 hours and

rural areas after 36 hours of interruption in electricity distribution (Nurmi et al., 2019). Thus they require a large amount of workforce to fix caused damages rapidly.

As Ulbrich et al. (2009) describe, there is no scientific consensus on how the occurrence and magnitude of extratropical storms will evolve in the future. Based on existing literature, the windstorm-related damages are increasing, while it remains unclear whether this is due to the higher exposure of society or the number and intensity of extratropical storms. Gregow et al. (2017) discovered that windstorm damages had increased significantly during the previous three decades, especially in northern, central, and western Europe. Also, several other studies suggest an increase in wind-related damages in Europe

(Csilléry et al. (2017); Haarsma et al. (2013); Gardiner et al. (2010)). Interestingly, some studies detected a decrease in the total number of extratropical storms (i.e. Donat et al. (2011)), while others found an increase in the number of extreme storms in specific regions, like western Europe and northeast Atlantic (Pinto et al., 2013). Another supporting view of a potential increase in extratropical storms in northern Europe can be found in the IPCC (2018) report. The report states that extratropical storm tracks are shifting towards the poles, which might affect the storminess in northern Europe. Thus, it may be concluded that

also the losses related to extratropical storms are likely to increase especially in northern Europe. However, as Barredo (2010) emphasizes, the cause for increased losses can at least partly be explained by the increasing exposure of society rather than the increased number of windstorms.

    Several previous studies respond to the demand for storm impact estimation for power distribution, many of them focusing on the hurricane-induced power blackouts in northern America (Eskandarpour and Khodaei (2017); Guikema et al. (2014, 2010);

Nateghi et al. (2014); Han et al. (2009); Wang et al. (2017); Allen et al. (2014); Chen and Kezunovic (2016); He et al. (2017); Liu et al. (2018)). Convective thunderstorms have also been investigated thoroughly. Li et al. (2015) introduced an area-based outage prediction method further developed to take power grid topology into account (Singhee and Wang, 2017). Shield et al. (2018) studied outage prediction by applying a random forest classifier to weather forecast data in a regular grid. Kankanala et al. used data from ground observation stations and experimented regression (Kankanala et al., 2011), a multilayer perceptron

neural network (Kankanala et al., 2012), and ensemble learning (Kankanala et al., 2014) to predict outages caused by wind and thunder. The Bayesian outage probability (BOP) prediction model developed by Yue et al. (2018) combines weather radar data and unifies it to a regular grid. Cintineo et al. (2014) create spatial objects from satellite and weather radar data, and track and classify the objects with the Naïve Bayesian classifier. Rossi (2015) developed a method to detect and track convective storms. The method was further developed to predict power outages (Tervo et al., 2019).

While much work exists on damage caused by hurricanes and convective thunderstorms, relatively few examples exist relating to outages caused by mid-latitude extratropical storms differing from hurricanes and convective storms in available data, time-span, and applicable methods for detecting and tracking. Extratropical storms are considered, for example, in Yang et al. (2020), where different decision tree methods are applied to a regular grid in the outage prediction task. Cerrai et al. (2019) also uses decision trees and regular grid for the outage prediction taking tree-leaf conditions into account as a predictive

feature. Related forest damage studies have been conducted with random forest classifiers and neural networks. Hart et al. (2019) showed that random forest regression and artificial neural networks could predict the number of falling trees in France caused by the wind. Hanewinkel (2005) conducted a similar study in Germany using artificial neural networks. Artificial

neural networks have been used to predict extreme weather in Finland (Ukkonen et al., 2017; Ukkonen and Mäkelä, 2019). The framework of IPCC (2018) emphasizes that the impacts of extreme weather risks can be analyzed by estimating the hazard, vulnerability, and exposure. In an increasing manner, connecting these fields (i.e., the natural hazard with the societal factors) is done with machine learning (Chen et al., 2008).

We present a novel method to identify, track, and classify extratropical storm objects based on how much power outages they are expected to induce. We adapt convective storm object detection (Rossi (2015), Tervo et al. (2019), Cintineo et al. (2014)) to find potentially harmful areas from extratropical storms by contouring objects from pressure and wind gust fields. Instead of highly-localized convective storms, we aim at larger but still regional geospatial accuracy so that, for example, damages in western and eastern Finland can be distinguished. We train a supervised machine learning model to classify storm objects according to their damage potential. To our knowledge, our method is the first that employs the extratropical storm objects as polygons and combines them with meteorological and non-meteorological features to predict power outages. The method can be used as a decision support tool in power distribution companies or as part of elaborating impact forecast by duty forecasters in national hydro-meteorological centers. The ERA5 atmospheric reanalysis (European Centre for Medium-Range Weather Forecasts, 2017) provides the primary meteorological input data for this study, while the national forest inventory provided by The Natural Resources Institute Finland (Luke) is used to represent the forest conditions in the prediction. Finally, historically occurred power outages from two sources are used to train the model. However, the operational use of the model would require the use of weather prediction data instead of reanalysis.

This paper is organized as follows: Chapter 2 presents the used data, which is followed by a step-by-step method description in Chapter 3. Chapter 3.1 discusses identifying storm objects and explains the storm tracking algorithm. Chapter 3.2 considers storm and forest characteristics, hereafter called features. Chapter 3.3 discusses how to define labels of storm objects based on the outage data. Chapter 3.4 describes the used machine learning methods. In Chapter 4, we discuss the performance of the method. Finally, Chapter 5 includes a discussion and conclusions.

## 2 Data

We base our method on three main data sources: ERA5 reanalysis data (Hersbach et al., 2019), multi-source national forest inventory (ms-nfi) provided by The Natural Resources Institute Finland (Luke), and occurred power outages obtained from two sources: First, the *local dataset* is gathered from two power distribution companies, Loiste and Järvi-Suomen Energia (JSE), located in Eastern Finland. Second, the *national dataset* is obtained from Finnish Energy (ET), a branch organization for the industrial and labor market policy of the energy sector. All data consider years from 2010 to 2018.

ERA5 is the newest generation reanalysis data provided by ECMWF. ERA5 covers the years from 1979 onward with a one-hour temporal resolution, has a horizontal resolution of 31 km, and covers the atmosphere using 137 levels up to a height of 80 km (Hersbach et al., 2019). Compared to in-situ wind observations, reanalysis data provides a spatiotemporally wider dataset. However, a question may arise about the accuracy of the reanalysis data. Ramon et al. (2019) examined the wind speed characteristics of a total of five state-of-the-art global reanalyses concerning 77 instrumented towers. In their study, ERA5

had the best agreement with in-situ observations on daily time scales; this suggests the ERA5 wind parameters to be adequate in windstorm damage examinations as well. ERA5 data are also known to contain unrealistically large surface wind speeds in some locations (European Centre for Medium-Range Weather Forecasts, 2019). None of these locations are, nevertheless, inside the geographical domain of this work.

The multi-source forest inventory data is based on field measurements, satellite observations, digital maps, and other geo-referenced data sources (Mäkisara et al., 2016). The data consists of estimates for the forest age, tree species dominance, the mean and total volume, and the biomass (total and tree species-specific). The original geospatial resolution of the data is 16 meters, which has been reduced to approximately 1.6 km resolution to speed up the processing. Taking into account the size of extratropical cyclones (diameter 1000 km) and the wide areas where wind damages typically occur e.g. near to the cold front,

we consider a resolution of 1.6 km being sufficiently high for modeling wind storm damages.

    Power outage data are obtained from two complementary sources. *The national dataset* is acquired from the Finnish Energy (2010-2018) who aggregates the data from power distribution companies in Finland. The national data are provided only for research purposes and for areas containing a minimum of six grid companies; this is, for example, to ensure energy users' anonymity. Therefore, the national dataset does not include exact locations of the faults. We have also obtained some parts of

105 the data with better spatial accuracy from two individual power distribution companies. In this paper, we refer to this data as *the local dataset*. In the local dataset, the fault locations are reported in relation to transformers, i.e. the spatial resolution of the outages ranges from a few meters to kilometers.

    Figure 1 illustrates the geographical coverage of the power outage data. The local dataset contains all outages from 2010 to 2018 in the northern area (Loiste) and outages related to major storms in the southern area (JSE), shown in Figure 1a. The

110 national dataset contains all outages in Finland from 2010 to 2018 divided into five regions, shown in Figure 1b. The national dataset contains in total 6 140 434 outages with relatively low geographical accuracy. On the other hand, the local dataset represents a substantially smaller geographical area with a good geographical accuracy but contains only 22 028 outages in total. We train our classification models, described in more detail in Chapter 3.4, with both datasets to evaluate their performance for different types of data.

**3 Method**

We predict power outages by classifying storm objects identified from gridded weather data into three classes based on the number of power outages the storm typically causes. The overall process consists of the following steps: (1) identifying storm objects from weather fields by finding contour lines of particular thresholds, (2) tracking the storm object movement, (3) gathering features of the storm objects, and (4) classifying each storm object individually. The classification is conducted for

each storm object separately to distinguish the different damage potential. Tracking is, however, necessary to gather necessary features such as object movement speed and direction. In the following, we discuss these phases in more detail.

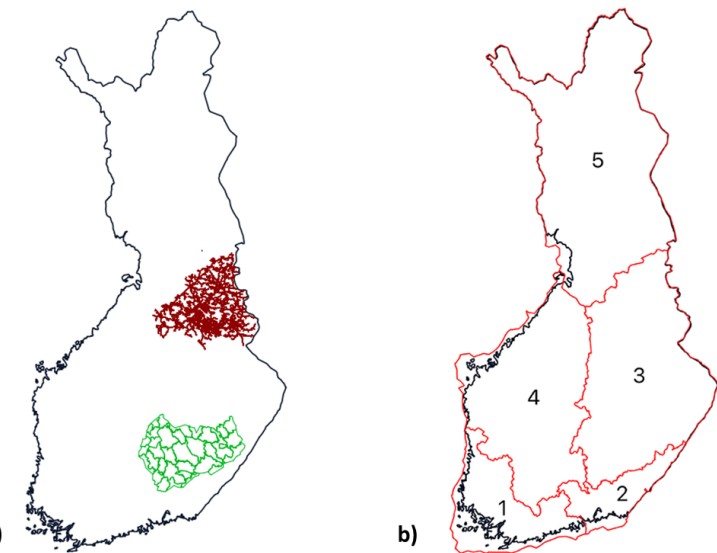

**Figure 1.** a) Geographical coverage of the outage data (local dataset). The red lines represent the power grid of Loiste (northern grid company) and the green lines the operative areas of JSE (southern grid company). Outages of the local dataset are collected from both areas. b) Regions in the national outage dataset. Outages are gathered from entire Finland and aggregated to the regions shown in the figure.

### 3.1 Identifying and tracking storm objects

Storm objects are identified by finding contour lines of 10-meter wind gust fields using $15\,\mathrm{m\,s^{-1}}$ thresholds from the ERA5 surface level grid with a time step of 1 hour. The contouring algorithm is capable of finding interior rings of the polygons. The

125 used wind gust fields did not, however, contain such cases. Thus one storm object represents a solid area (polygon) where the hourly maximum wind gust exceeds $15\,\mathrm{m\,s^{-1}}$ during one particular hour. The threshold of $15\,\mathrm{m\,s^{-1}}$ is selected as different sources indicate Finland being vulnerable for windstorms and rather moderate winds (from $15\,\mathrm{m\,s^{-1}}$) causing damages to forests (Valta et al., 2019; Gardiner et al., 2013). Valta et al. (2019) developed a method to estimate the windstorm impacts on forests by combining the recorded forest damages from the nine most intense storms and their observed maximum inland wind

gusts. According to the formula developed in the study, the inland wind gusts of $15\,\mathrm{m\,s^{-1}}$ alone result in forest damages of $1800\,\mathrm{m^3}$. We also identify pressure objects by finding contour lines using a $1000\,\mathrm{hPa}$ threshold to connect potentially distant storm objects around the low-pressure center to the same storm event.

After identification, storm objects are tracked by connecting them with each other. Each storm object is first connected to nearby pressure objects from the current and preceding time steps. If pressure objects do not exist within the distance threshold,

the object is connected to nearby storm objects from the current and preceding time steps. The Algorithm enables assigning each storm object to an overall event (low pressure system) and tracking the objects' movement. Algorithm 1 shows the details of the process.

We use a 500 km distance threshold for the distance between the storm and pressure objects. As the typical diameter of an extratropical storm is approximately 1000 km (Govorushko, 2011), we assume the damaging storm objects to situate a maximum 500 km from the center of the low pressure. The threshold for movement speed is 200 $\mathrm{km\ h^{-1}}$ for storm objects and 45 $\mathrm{km\ h^{-1}}$ for pressure objects. In other words, storm objects are not assumed to move more than 200 km and pressure objects more than 45 km from the preceding hourly time step (Govorushko, 2011). Convective storms may move faster but are outside the focus of this work.

## 3.2 Extracting storm object features

We characterize the storm objects identified by the methods discussed in Section 3.1 using the features listed in Table 1. The features are structured as four groups. The first group is a number of object characteristics such as size and movement speed and direction, which are calculated from the contoured storm objects themselves. As the second group, relevant weather conditions, such as wind speed, temperature, and others, are extracted from ERA5 data. We aggregate values as a minimum, maximum, average, and standard deviation calculated over all grid cells under the object coverage to represent each parameter with one number. Third, as most of the outages are caused by the trees falling on power grid lines (Campbell and Lowry, 2012), the characteristics of the forest contribute to the damages (Peltola et al., 1999), we complement our data with forest information. As for weather parameters, values are aggregated over the storm object coverage. The fourth group consists of the number of outages and affected customers used as labels in the model training process discussed in more detail in Chapter 3.4.

We selected the 35 parameters based on two main criteria: First, we prepared a list of potential parameters detected in related studies, e.g. Suvanto et al. (2016); Peltola et al. (1999); Valta et al. (2019), or identified through the empirical experience of duty forecasters (Weather and Safety Center of Finnish Meteorological Institute - Duty forecasters, 05/2020). Second, we selected the relevant parameters, which were available to us or accessible with a reasonable effort. However, some possibly essential parameters, like soil temperature from ERA5 reanalysis, were left out because of the slow downloading process.

After the preliminary selection of the parameters, we conducted dozens of light experiments using different combinations of parameters and models to find the best possible setup. To this end, we fitted the Gaussian distribution to each parameter using at first all samples, then samples with few outages, and finally with many outages (classes 1 and 2 specified in Section 3.3). While many other distributions are known to suit better in modeling particular parameters, such as Gamma in precipitation, Weibull in wind speed, and Lognormal in cloud properties (Wilks, 2011), the Gaussian distribution is a sufficient simplification to help in selecting relevant parameters. We visually inspected the differences between fitted Gaussian distributions to deduce the potential relevance of the parameter. Supposedly the distribution of one parameter is different for all samples and samples with many outages, and the classification method may exploit the parameter to predict the damage potential of the storm object. The distributions of some selected parameters are shown in Appendix A. In total, 35 parameters, shown as boldfaced in Table 1, were chosen for the final classification.

---

**Algorithm 1** Storm tracking

---

**Input**

  Storm and pressure objects $S_o$ arranged by time

  *pressure distance threshold*

  *wind distance threshold*

  *speed threshold*

  *time step*

**Output**

  Connected storm and pressure objects with storm $ID$

**for all** storm and pressure object $O_{w|p} \in S_o$ **do**

  *current time* ← time of the object $O_{w|p}$

  *previous time* ← *current time* − *time step*

  Current time pressure objects $S_p^c$ ← pressure objects having centroid within *pressure distance threshold* from
         object $O_{w|p}$ centroid and time stamp *current time*

  Previous time pressure objects $S_p^p$ ← pressure objects having centroid within *speed threshold* from
         object $O_{w|p}$ centroid and time stamp *previous time*

  Current time storm objects $S_w^c$ ← storm objects having centroid within *wind distance threshold* from
         object $O_{w|p}$ centroid and time stamp *current time*

  Previous time storm objects $S_w^p$ ← storm objects having centroid within *speed threshold* from
         object $O_{w|p}$ centroid and time stamp *previous time*

  **if** pressure object $O_p^c \in S_p^c$ exists with $ID$ **then**

    Use pressure object $O_p^c$ $ID$

  **else if** pressure object $O_p^p \in S_p^p$ exists with $ID$ **then**

    Use previous time pressure object $O_p^p$ $ID$

  **else if** storm object $O_w^c \in S_w^c$ exists with $ID$ **then**

    Use storm object $O_w^c$ $ID$

  **else if** storm object $O_w^p \in S_w^p$ exists with $ID$ **then**

    Use previous time storm object $O_w^p$ $ID$

  **else if** storm or pressure object $O_{w|p}^p \in S_w^p \cup S_p^p$ exists without $ID$ **then**

    Give new $ID$ to the previous object $O_{w|p}^p$ and current object $O_{w|p}$

  **else**

    Leave object $O_{w|p}^p$ without $ID$

  **end if**

**end for**

---

**Table 1.** Extracted features. Features used in the final classification marked as bold.

| Feature | Aggregation | Explanation |
|---|---|---|
| **Speed** | - | Object movement speed |
| **Angle** | - | Object movement angle |
| **Area** | - | Object size |
| **Area difference** | - | Object area difference to the previous time step |
| **Week** | - | Week of the year |
| Snowdepth | average, minimum, maximum | Snow depth |
| **Total column water vapor** | **average, minimum, maximum** | Total amount of water vapour |
| **Temperature** | **average, minimum, maximum** | 2 meter air temperature |
| Snowfall | average, minimum, maximum, sum | Snowfall (meter of water equivalent) |
| Total cloud cover | average, minimum, maximum | Total cloud cover (0-1) |
| **CAPE** | **average**, minimum, **maximum** | Convective available potential energy (J/kg) |
| Precipitation kg/m2 | average, minimum, maximum, sum | Precipitation amount (kg/m2) |
| **Wind gust** | **average**, minimum, **maximum, standard deviation** | Hourly maximum wind gust (m s$^{-1}$) |
| **Wind Speed** | **average**, minimum, **maximum, standard deviation** | 10 meter wind speed (m s$^{-1}$) |
| **Wind Direction** | **average**, minimum, maximum, **standard deviation** | Wind direction (degrees)) |
| **Dewpoint** | **average, minimum, maximum** | Dewpoint) |
| **Mixed layer height** | **average, minimum, maximum** | Boundary layer height |
| **Pressure** | average, **minimum**, maximum | Air pressure |
| **Forest age** | **average**, minimum, maximum, standard deviation | The age of the growing stock on a forest stand |
| **Forest site fertility** | **average**, minimum, maximum, standard deviation | Group of the forest by vegetation zones |
| Forest stand mean diameter | **average**, minimum, maximum, standard deviation | Forest stand mean mean diameter |
| **Forest stand mean height** | **average**, minimum, maximum, standard deviation | Forest stand mean height |
| **Forest canopy cover** | **average**, minimum, maximum, standard deviation | Forest canopy cover fraction (0-100%) |
| Outages | - | Number of occured outages |
| Customers | - | Number of affected customers |
| Tansformers | - | Number of transformers under the object |
| All customers | - | Number of customers under the object |
| **Class** | - | Assigned class |

### 3.3 Defining classes

As shown in Figures 2a and 2b, the outages in the local dataset are concentrated heavily on 'hot-spots', assumingly, due to forest characteristics and network topology. The local dataset contains 24 542 storm objects and 5 837 outages connected to 2 363 storm objects. Thus 22 179 storm objects in the local dataset did not cause any outages. The local power outage data contain 16 191 outages, which can not be connected to any storm object. The national dataset contains 142 873 storm objects and 5 965 324 outages connected to 33 796 storm objects. 109 077 storm objects are not connected to any outages, and 175 110

outages can not be connected to any storm object.

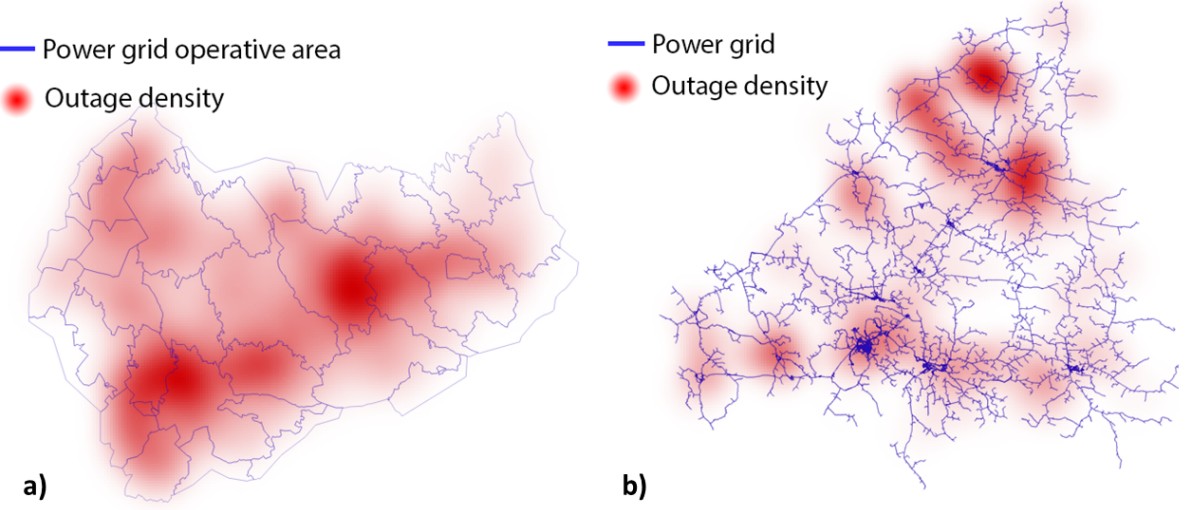

**Figure 2.** Spatial distribution of the outages between 2010 and 2018 visualised as a spatial heatmap. a) JSE network (southern area) b) Loiste network (northern area)

  It should be noticed that the damage may occur anywhere in the power grid. Outages are, however, always reported as transformers without electricity. Typically one physical damage between the transformers causes several transformers to lose power. Power grid operators can often turn part of the transformers back to operation even before fixing the actual damage, which causes an unavoidable noise to the datasets.

Figure 3 represents the number of outages and storm objects in both local and national datasets. We can identify a large amount of 15 m s$^{-1}$ storm objects in both sets, indicating that moderate wind without other influencing factors does not damage the transformers. When identifying storm objects with the contour of 20 and 25 m s$^{-1}$, the number of objects reduces and starts to correlate more with a high number of outages, which supports views of previous studies showing the significance of stronger wind gusts to more severe storm damages. The method seems to identify also the most critical storm days by

capturing several storm objects for those days. For instance, at the end of 2013, when the three major storms Eino, Oskari,

Seija (Valta et al., 2019) hit Finland, both datasets contain plenty of storm objects with the $20 \ \mathrm{m\ s^{-1}}$ threshold. Nevertheless, our experiments indicated that employing $15 \ \mathrm{m\ s^{-1}}$ storm objects yielded the best results. This is described more in Chapter 4.

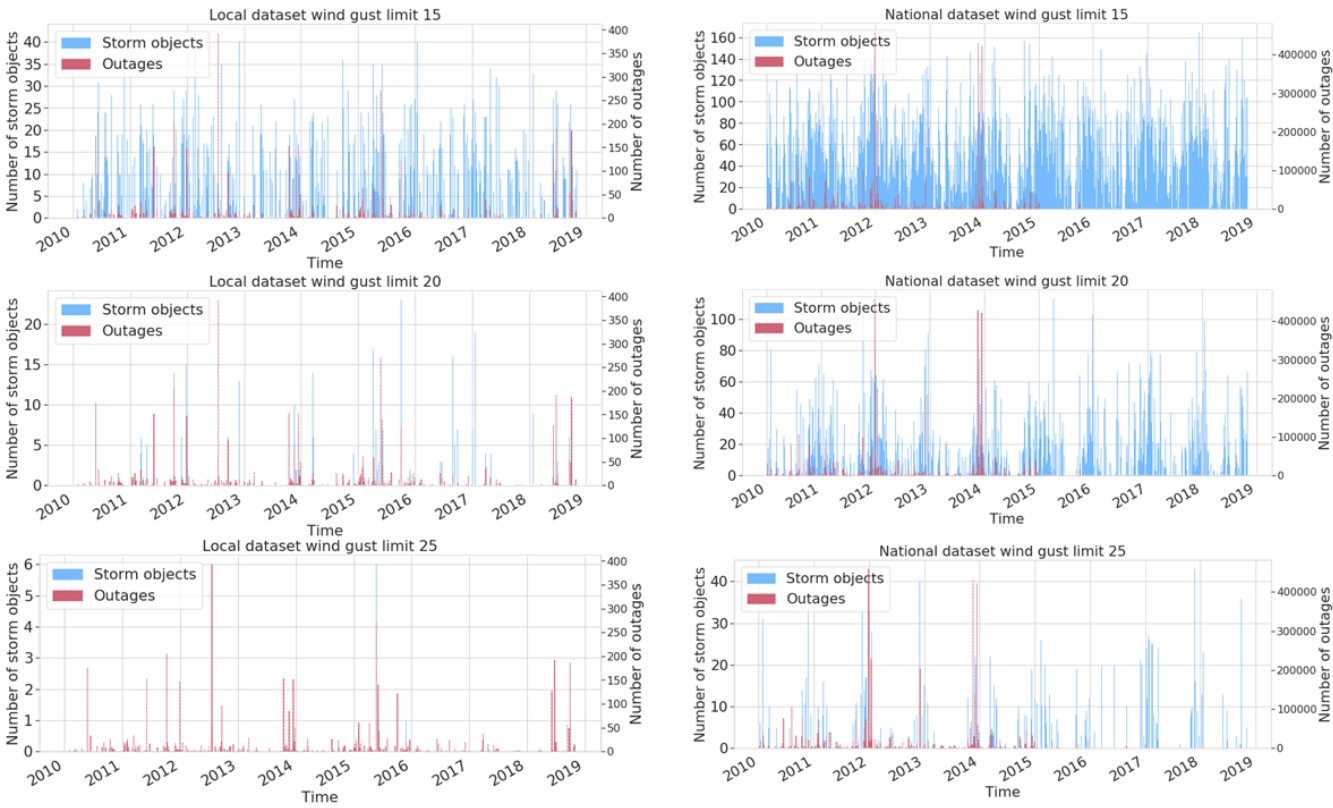

**Figure 3.** Storm object time series (15, 20 and 25 $\mathrm{m\ s^{-1}}$ contours) with occurred outages for local and national datasets.

Figure 4 illustrates how much outages a single storm object typically produces. In the local dataset, most of the storm objects cause only a few outages. Only 65 storm objects, which are only 0.3 percent of the whole dataset, induced more than ten outages. On the other hand, in the national dataset where one storm object typically affects several different transformers, 17 587 storm objects have caused more than ten outages, representing 12 percent of the whole dataset. Figure 5 renders how many customers are typically affected by one outage. The figure contains all outages in both datasets, whether they are related to a storm or not. In the local dataset, usually 20-30 customers lose electricity in one outage. In the national dataset, only six customers usually lose electricity in one outage. We assume that this roots to different network topologies between the areas. Notably, in some rare cases, a much higher number of customers are affected. We assume that these cases occur typically in urban areas and are rare because the power network is mainly underground in these areas.

We use three classes designed together with power grid companies aiming at a simple "at glance" view for power grid operators. Class 0 represents no damage, class 1 low damage, and class 2 high damage. As the number of outages produced by a single storm object varies significantly in the local and national datasets, we decided to define separate limits for the local

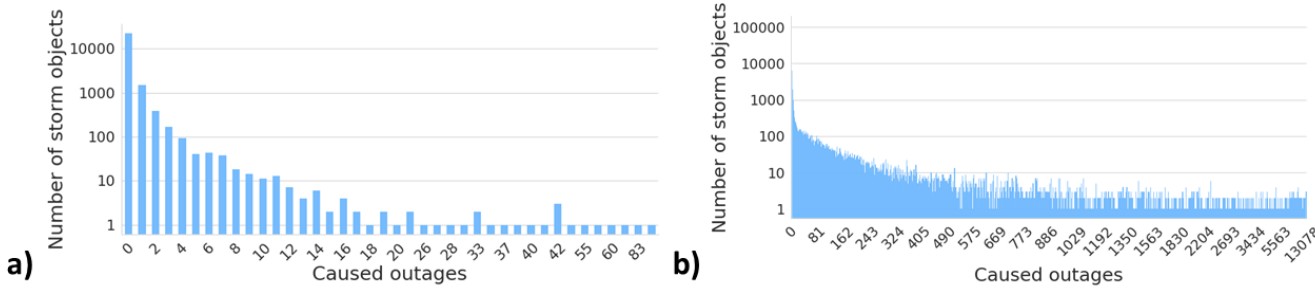

**Figure 4.** Number of storm objects per caused outages in a) local dataset b) national dataset.

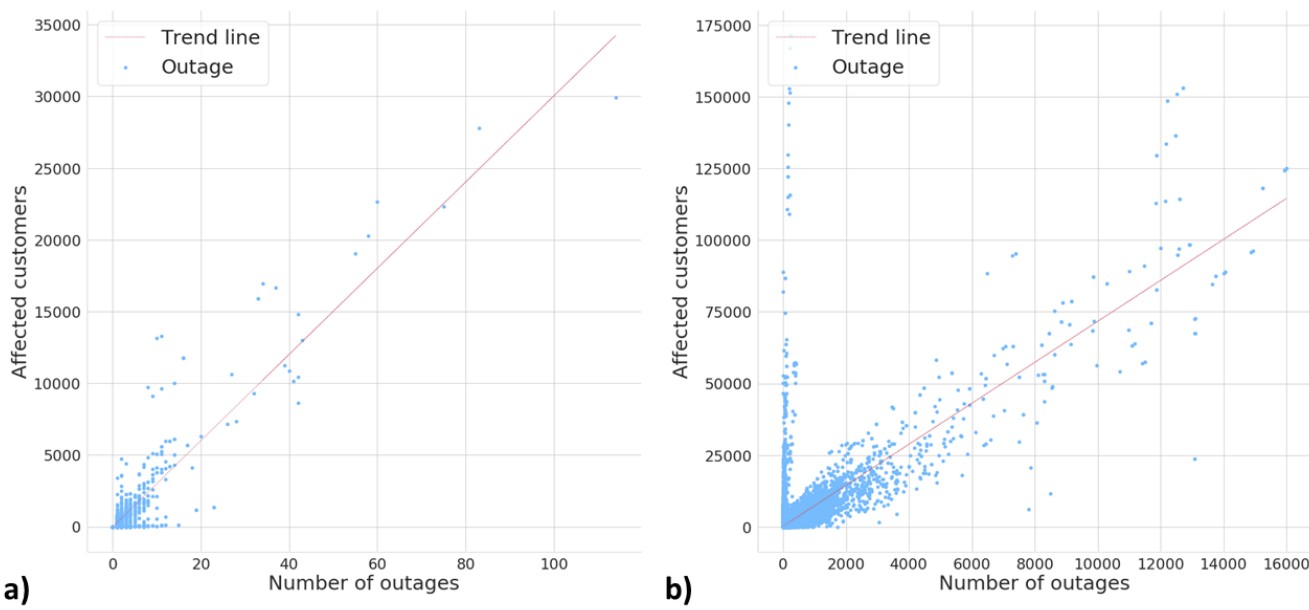

**Figure 5.** Relationship between number of outages and affected customers in a) local dataset and b) national dataset.

and the national datasets. The detailed limits are listed in Table 2. Class 1 is defined such that it represents roughly 80 percent of all cases with at least one outage. Class sizes are highly imbalanced as most of the storm objects do not cause any damage.

### 3.4 Classifying storm objects

We centered and normalized the data points by substracting the empirical mean and then dividing it by the empirical standard deviation. The hyperparameters were determined using random search 5-fold cross-validation (Bergstra and Bengio, 2012). To cope with the imbalanced class distribution, we generate artificial training samples using the synthetic minority over-sampling

**Table 2.** Class definitions

| Class | Outage limit in local dataset | Local dataset size | Outage limit in national dataset | National dataset size |
|-------|------------------------------|--------------------|---------------------------------|----------------------|
| **0** | 0 | 5 624 | 0 | 76 215 |
| **1** | 1-3 | 353 | 1- 140 | 14 417 |
| **2** | $\geq 4$ | 181 | $\geq 141$ | 3 085 |

technique SMOTE (Chawla et al., 2002). The SMOTE creates new training samples based on their $k = 5$ nearest neighbors following:

$$x_{new} = x_i + \lambda \times (x_{zi} - x_i) \tag{1}$$

where $x_i$ is an original class sample, $x_{zi}$ is one of $x_i$'s $k$ nearest neighbor and $\lambda$ is a random variable drawn uniformly from the interval $[0, 1]$. After augmentation, all classes have an equal number of samples, which reduces the tendency of classification methods to always predict the majority class.

Five different models were evaluated to classify storm objects. We omit the mathematical definitions but shortly discuss the characteristics of different models and describe the implementation details chosen in this work.

**Random forest classification (RFC)** is based on a random ensemble of decision trees and aggregating results from individual trees to the final estimate. Trees in the ensemble are constructed with four steps: 1) use bootstrapping to generate a random sample of the data, 2) randomly select a subset of features at each node, 3) determine the best split at the node using loss function, 4) grow the full tree (Breiman, 2001). RFC is good to cope with high-dimensional data. It has also been found to provide adequate performance with imbalanced data (Tervo et al., 2019; Brown and Mues, 2012) and is widely used with weather data (e.g. Karthick et al. (2020); Cerrai et al. (2019); Lagerquist et al. (2017)). The method is prone to overfit, which is why hyperparameter-tuning is very important. Hyperparameters used in this work are listed in Table 3. We use RFC with the Gini impurity loss function.

**Table 3.** Hyperparameters for the RFC

| Parameter | Value |
|-----------|-------|
| Number of trees in the forest | 500 |
| Max depth | unlimited |
| Minimum nr. of samples to split | 2 |
| Minimun nr of samples to leaf | 1 |
| Features to consider for split | $\sqrt{\text{num. of feat.}}$ |
| Max nro of leaf nodes | unlimited |

**Support Vector Classifiers (SVC)** construct a hyper-plane or classification function in a high-dimensional feature space and maximize a distance between training samples and the hyperplane. The hyper-planes may be constructed with nonlinear kernels such as gaussian radial basis function (RBF) (Shawe-Taylor et al., 2004) that often reform a nonlinear classification problem to a linear one. Operating in the high-dimensional feature space without additional computational complexity makes SVC an attractive choice to extract meaningful features from a high-dimensional dataset. A domain-specific expert knowledge can also be capitalized on the kernel design. On the other hand, finding the correct kernel is often a difficult task. Training SVC is a convex optimization problem meaning that it has no local minima. Depending on the kernel, a training process may, however, be a very memory-intensive process.

Suppose the SVM output is assumed to be the log odds of a positive sample. In that case, one can fit a parametric model to obtain the posterior probability function and thus get probabilities for samples to belong to the particular class (Platt et al., 1999). For more details, we request the reader to consult for example Chang and Lin (2011) and Platt et al. (1999).

We implement the SVC in two phases. First, we separate class 0 (no outages) and other samples employing SVC with radial basis function (RBF), defined in Equation 2. Second, we distinguish classes 1 and 2 using SVC with a dot-product kernel defined in Equation 3 (Williams and Rasmussen, 2006). The second phase is performed only for the samples predicted to cause outages in the first phase. The approach is similar to the often-used one-vs-one classification, where a binary classifier is fitted for each pair of classes. In our case different kernels were used for different pairs.

$$k_{RBF}(\mathbf{x}, \mathbf{x}') = \exp\left(-\gamma ||\mathbf{x} - \mathbf{x}'||^2\right) \tag{2}$$

where $\mathbf{x}$ and $\mathbf{x}'$ are two samples in the input space and $\gamma$ is a kernel coefficient parameter.

$$k_{\cdot}(\mathbf{x}, \mathbf{x}') = \sigma_0 + x \cdot x' \tag{3}$$

where $\mathbf{x}$ and $\mathbf{x}'$ are two samples in the input space and $\sigma$ is a kernel inhomogenity parameter.

**Gaussian Naive Bayes (GNB)** (Chan et al., 1979) is a well-known and widely used method based on the Bayesian probability theory. The method assumes that all samples are independent and identically distributed (i.i.d), which does not naturally hold for the weather data. Despite the internal structure of the data, GNB is still used for weather data (e.g. Kossin and Sitkowski (2009); Cintineo et al. (2014); Karthick et al. (2020)) and worth investigating in this context. The classification rule in GNB is $\hat{y} = \arg\max_y P(y) \prod_{i=1}^n P(x_i \mid y)$, where $P(y)$ is a frequency of class $y$ and $P(x_i \mid y)$ is a likelihood of the $i$th feature assumed to be gaussian. Because of the naive i.i.d assumption, each likelihood can be estimated separately, which helps to cope with a curse of dimensionality and enable GNB to work relatively well with small datasets. On the other hand, estimating likelihoods can be done effectively and iteratively, enabling the GNB to scale to large datasets. As a downside, the simple method may lack expression power to perform well in a complex context.

**Gaussian Processes (GP)** (Rasmussen, 2003) is a non-parametric probabilistic method that interprets the observed data points as realizations of a Gaussian random process. GP is widely used for example in weather observation interpolation

*kriging* (Holdaway, 1996). GP is a very flexible and powerful but computationally expensive method, which tends to lose its power with high-dimensional data. GP hinges on a kernel function that encodes the covariance between different data points. As a kernel, we use a product of dot-product kernel (Equation 3) and pairwise kernel with laplacian distance (Rupp, 2015), defined in Equation 4. The kernel parameters were optimized on the training data by maximizing the log-marginal-likelihood.

$$k_{pairwise}(\mathbf{x}, \mathbf{x}') = \exp\left(-\gamma ||\mathbf{x} - \mathbf{x}'||_1\right) \tag{4}$$

where $\mathbf{x}$ and $\mathbf{x}'$ are two samples in the input space and $\gamma$ is a kernel coefficient parameter.

**Multilayer perceptrons (MLP)** (Goodfellow et al., 2016) are the most basic form of artificial neural networks. Good results achieved by MLP in predicting storms (Ukkonen and Mäkelä, 2019), they are a natural choice to experiment in this work. Neural networks are very adaptive methods as they can learn a representation of the input at their hidden layers. Unlike GNB, they do not make any assumptions about the distribution of the data. As a downside, MLP requires large amounts of data, and the training process is computing-intensive. They also have a large number of hyperparameters to be optimized, including the correct network topology.

We searched the correct model parameters and network topology for local and national datasets by running multiple iterations of random search 5-fold cross-validation to obtain the best possible micro average of F1-score (defined in Chapter 4) employing Talos library (Autonomio, 2020). The final setup composes of Nadam optimizer (Dozat, 2016), random normal initializer, and relu activation function for hidden layers. Binary cross-entropy was used as a loss function. Optimal network topology varied in different datasets: For the local dataset, the best results were obtained with a network containing three hidden layers with 75, 145, and 35 neurons. For the national dataset, the best results were obtained with a network containing three hidden layers with 75, 195, and 300 neurons. During the optimization process, the results varied between different setups from 0.6 to 0.95 in terms of F1-score.

## 4 Results

We used two different methods for splitting the data into training and test set. The first method is to use 25 percent of randomly picked samples in the test set. The second method is to construct a test set from a one-year continuous time range (2010-2011). Both approaches have their advantages. Continuous time range ensures that the model has not seen any autocorrelated samples caused by an internal structure of the weather data in the training phase (Roberts et al., 2017). However, having only nine years of data from a relatively small geographical area, the continuous test set cannot contain many storms as most of the data needs to be reserved for the training process. Thus, the test set may only contain a single type of storms to which the model may work especially well or bad. Picking the test set randomly minimizes this risk and provides more insight into the model performance.

We evaluate the models with a weighted average of precision and recall, and both weighted and macro average of F1-score. Precision (Equation 5) reports how many samples are correctly predicted to belong to a class. Recall (Equation 6) tells how many samples belonging to a class are found in the prediction. F1-score (Equations 7 and 8) calculates a harmonic mean of

precision and recall. Finally, as the datasets are extremely imbalanced, we calculate a weighted average of the metrics utilizing a number of samples in each class and a macro average of F1-score using an average of F1-score of each class. A model with a higher macro average of F1-score performs better with small classes. The selected metrics do not take a distance between predicted and true class into account. It is naturally worse to predict, for example, class 0 (no damage) in the case of true class 2 (high damage) than in the case of true class 1 (low damage). We decided, however, to use metrics that measure the method performance properly with imbalanced classes.

$$Precision = \frac{1}{\sum_{c \in \mathcal{C}} |\hat{y}_c|} \sum_{c \in \mathcal{C}} \left( |\hat{y}_c| \frac{tp}{tp + fp} \right) \tag{5}$$

where $\mathcal{C}$ represents the set of classes, $\hat{y}$ predicted the class, $tp$ true positives, and $fp$ false positives.

$$Recall = \frac{1}{\sum_{c \in \mathcal{C}} |\hat{y}_c|} \sum_{c \in \mathcal{C}} \left( |\hat{y}_c| \frac{tp}{tp + fn} \right) \tag{6}$$

where $\mathcal{C}$ represents the set of classes, $\hat{y}$ predicted the class, $tp$ true positives, and $fn$ false negatives.

$$F1_{weighted} = \frac{1}{\sum_{c \in \mathcal{C}} |\hat{y}_c|} \sum_{c \in \mathcal{C}} \left( |\hat{y}_c| \frac{Precision_c \times Recall_c}{Precision_c + Recall_c} \right) \tag{7}$$

where $\mathcal{C}$ represents the set of classes, $\hat{y}$ predicted the class, Precision defined in Equation 5, and Recall defined in Equation 6.

$$F1_{macro} = \frac{1}{|\mathcal{C}|} \sum_{c \in \mathcal{C}} \left( \frac{Precision_c \times Recall_c}{Precision_c + Recall_c} \right) \tag{8}$$

where $\mathcal{C}$ represents the set of classes, Precision defined in Equation 5, and Recall defined in Equation 6.

Tables 4 and 5 divulge the results for each models using the local and national dataset respectively. Models trained with the local dataset can reach the better-weighted F1-score, while the best models trained with the national dataset provide a significantly better macro average of F1-score. The national dataset contains many more samples in classes 1 and 2, which enables models to learn the classes better and thus enhance the macro average of the F1-score. Whether the test set is randomly chosen or continuous does not seem to make a large difference in most cases. The only affected model is the RFC having contradictory better results trained with the continuous test set from the local dataset and the random test set from the national dataset. Assumingly, this squeal more about the unstable performance of RFC than the relevance of the dataset split method.

The confusion matrices are depicted in Figure 6. RFC provides the best results in terms of the selected metrics. However, closer exploration reveals that this performance is largely due to the best performance in predicting class 0, which is the largest class. SVC results are one of the most balanced ones being the best only in the local dataset with a random test set but yielding good stable results in all cases. The confusion matrix, shown in Figure 6b, displays that it is not the best model to predict class

**Table 4.** Results for each models trained with the local dataset obtained from two local power grid companies (defined in Chapter 3.3)

| Model | Split method | Precision | Recall | Weighted F1-score | | Macro AVG F1-score | |
|---|---|---|---|---|---|---|---|
| | | test | test | train | test | train | test |
| **Random Forest Classifier (RFC)** | Random | 0.82 | 0.76 | 0.93 | 0.79 | 0.93 | 0.40 |
| | Continuous | 0.88 | **0.91** | 0.93 | **0.89** | 0.93 | **0.48** |
| **Support Vector Classifier (SVC)** | Random | 0.85 | 0.73 | 0.78 | 0.78 | 0.78 | **0.44** |
| | Continuous | 0.87 | 0.72 | 0.77 | 0.78 | 0.77 | 0.42 |
| **Gaussian Naive Bayes (GNB)** | Random | **0.87** | 0.61 | 0.59 | 0.70 | 0.59 | 0.42 |
| | Continuous | **0.89** | 0.59 | 0.59 | 0.69 | 0.59 | 0.40 |
| **Gaussian Processes (GP)** | Random | 0.84 | 0.70 | 1.0 | 0.76 | 1.0 | 0.43 |
| | Continuous | 0.85 | 0.67 | 0.94 | 0.74 | 0.94 | 0.41 |
| **Multilayer perceptor (MLP)** | Random | 0.82 | **0.81** | 0.98 | **0.80** | 0.91 | 0.41 |
| | Continuous | 0.81 | 0.79 | 0.97 | 0.80 | 0.91 | 0.41 |

**Table 5.** Results for each models trained with the national dataset covering whole Finland (defined in Chapter 3.3)

| Model | test set split method | Precision | Recall | Weighted F1-score | | Macro AVG F1-score | |
|---|---|---|---|---|---|---|---|
| | | test | test | train | test | train | test |
| **Random Forest Classifier (RFC)** | Random | **0.83** | **0.84** | 1.0 | **0.83** | 1.0 | **0.62** |
| | Continuous | **0.77** | **0.81** | 1.0 | **0.78** | 1.0 | 0.40 |
| **Support Vector Classifier (SVC)** | Random | 0.81 | 0.61 | 0.68 | 0.68 | 0.68 | 0.60 |
| | Continuous | 0.62 | 0.60 | 0.60 | 0.60 | 0.60 | 0.60 |
| **Gaussian Naive Bayes (GNB)** | Random | 0.75 | 0.60 | 0.66 | 0.66 | 0.45 | 0.39 |
| | Continuous | **0.77** | 0.60 | 0.45 | 0.66 | 0.45 | 0.40 |
| **Gaussian Processes (GP)** | Random | 0.57 | 0.56 | 0.71 | 0.55 | 0.71 | 0.55 |
| | Continuous | 0.67 | 0.65 | 0.94 | 0.65 | 0.94 | **0.61** |
| **Multilayer perceptor (MLP)** | Random | 0.79 | 0.75 | 0.94 | 0.77 | 0.90 | 0.52 |
| | Continuous | 0.76 | 0.78 | 0.93 | 0.78 | 0.85 | 0.40 |

0, but only a little share of true class 2 cases and the smallest share of true class 1 cases are predicted as class 0. That is to say, SVC misses the smallest number of destructive storms, although it confuses in the amount of caused damage.

GP is another strong option that performs even better with class 0 while still providing good performance with class 2. A significant connecting aspect between GP and SVC is an almost identical kernel. Based on these experiments, RBF and pairwise kernels separate harmless and harmful samples from each other while dot-product kernel separates the classes 1 and 2 even better than exponential functions. We select GP for further analysis in this paper since it provides the best performance in class 2.

Using the $15 \mathrm{~m~s}^{-1}$ threshold for detecting storm objects yields clearly better results than the $20 \mathrm{~m~s}^{-1}$ threshold. For example, SVC trained with the national dataset using the $20 \mathrm{~m~s}^{-1}$ threshold and randomly chosen test set provide only 0.48 macro average of F1-score being 12 percentage points below corresponding model using the $15 \mathrm{~m~s}^{-1}$ threshold. The $15 \mathrm{~m~s}^{-1}$ threshold have two major advantages compared to the $20 \mathrm{~m~s}^{-1}$. First, it provides a significantly larger dataset and second, in contrast to the $20 \mathrm{~m~s}^{-1}$ threshold, it is able to catch virtually all extratropical storms causing outages.

## 4.1 Feature importances in the model performance

The relevance of the individual predictive features can be explored by using the permutation test, as done by Breiman (2001). First, the baseline score of the fitted model is calculated using the test set. Then each feature is randomly permuted, and the difference in the scoring function is calculated. The random permutation is repeated 30 times for each parameter, and the average of the results is used. The procedure offers information on how important the feature is to obtain good results. It should be mentioned that highly correlated features may get low importance as other features work as a proxy to the permuted feature. However, using completely independent features is not possible in weather data since weather parameters are often dependent on each other, and eliminating even the most apparent pairs from the used features impaired the results in our experiments.

We used the macro average of F1 defined in Equation 8 as a scoring function and the randomly selected test set from the national data. The relevance is shown in Figure 7. Most features show at least little relevance for the results. The first twelve features are significantly more relevant than the rest. The most important features contain at least one representative of all meteorological parameters used in the training. In other words, all employed meteorological parameters are important for the prediction, while different aggregations are contributing to the "fine-tuning" of the model.

As Figure 7 shows, the most significant parameter regarding our model performance is the average wind speed. Numerous studies support our result of wind being the most important damaging factor (Virot et al., 2016; Valta et al., 2019; Jokinen et al., 2015). The studies are, however, highlighting the importance of maximum wind gusts instead of the average wind. Surprisingly, in our analysis, the wind gust speed does not belong to the most critical parameters. Instead, maximum mixed layer height, related to the wind gustiness, contributes crucially to the model performance. The dependencies between predictive features might be one reason for some parameters to have a lower rank in the results.

The stand mean diameter and height are the most important features regarding the forest parameters, which corresponds to our expectations. Previous studies also state these features to influence the wind damage in forests (Pellikka and Järvenpää, 2003) and hence indirectly electricity grids. As Pellikka and Järvenpää (2003) and Suvanto et al. (2016) discuss, also the age of the forest has an impact on storm damages. However, in the feature importance test, forest age does not seem to contribute significantly to the prediction outcome.

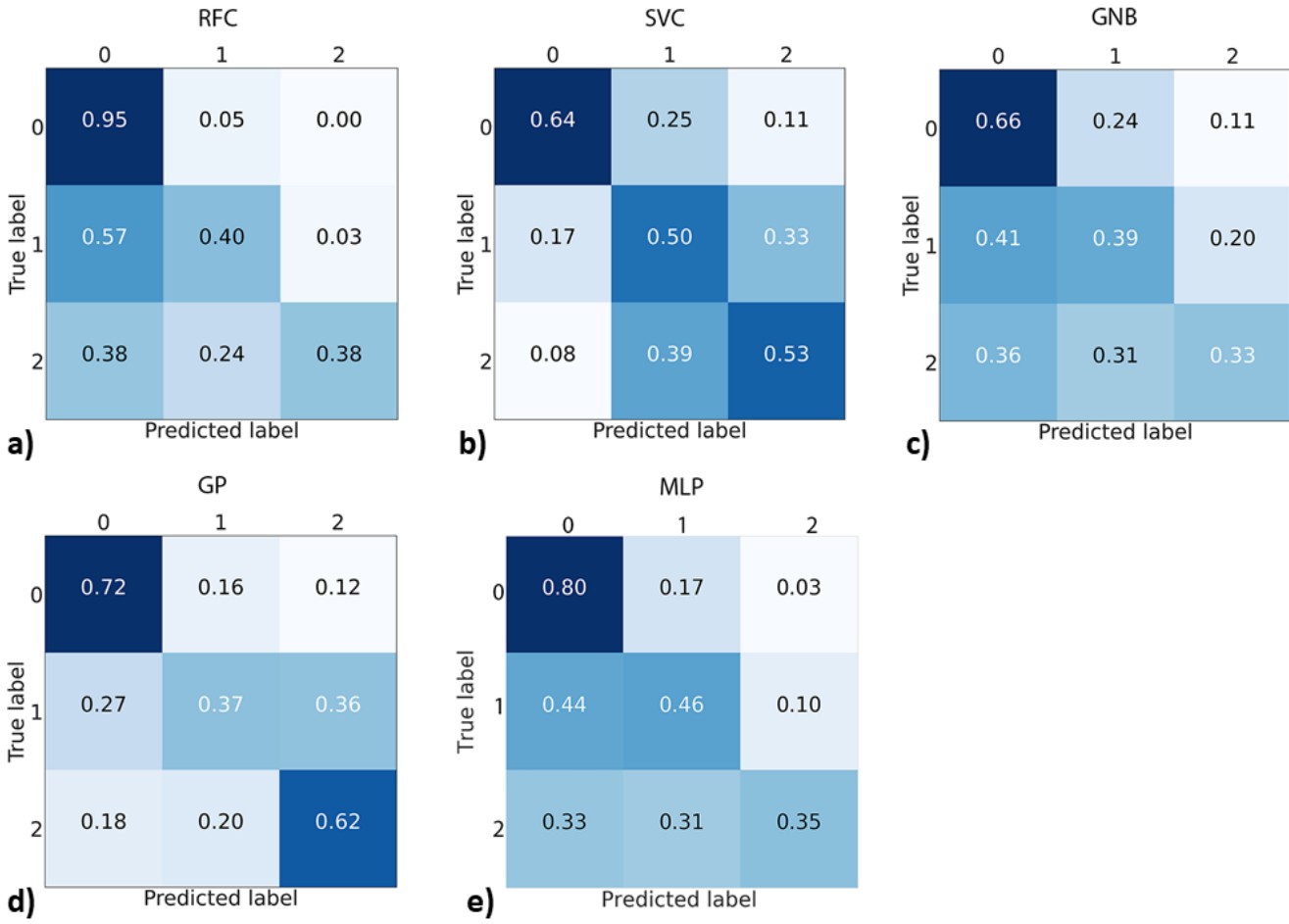

**Figure 6.** Confusion matrices produced using the randomly selected national dataset and a) RFC b) SVC c) GNB d) GP e) MLP. Each cell of the confusion matrices represents a share of predictions having a corresponding combination of predicted and true class. For example, the middle right cell tells the share of samples belonging to class 1 but predicted to have class 2.

The most important object feature is the size of the object. Object movement speed and direction did not contribute strongly to the results. However, previous studies indicate that besides the size of the impacted area, the duration of strong winds – i.e., the propagation speed of the system – influences also the amount of damage (Lamb and Knud, 1991).

## 4.2  Case Examples

We illustrate the prediction produced using GP classification method with the three most interesting examples of well-known storms in Figure 8. We chose the cases among a number of test cases to illustrate the strengths and weaknesses of the method. The examples are chosen from the randomly picked test set, which was not used to train the model. Because of the random sample, we cannot represent the entire prediction of individual storms, only individually picked time steps. In two of the

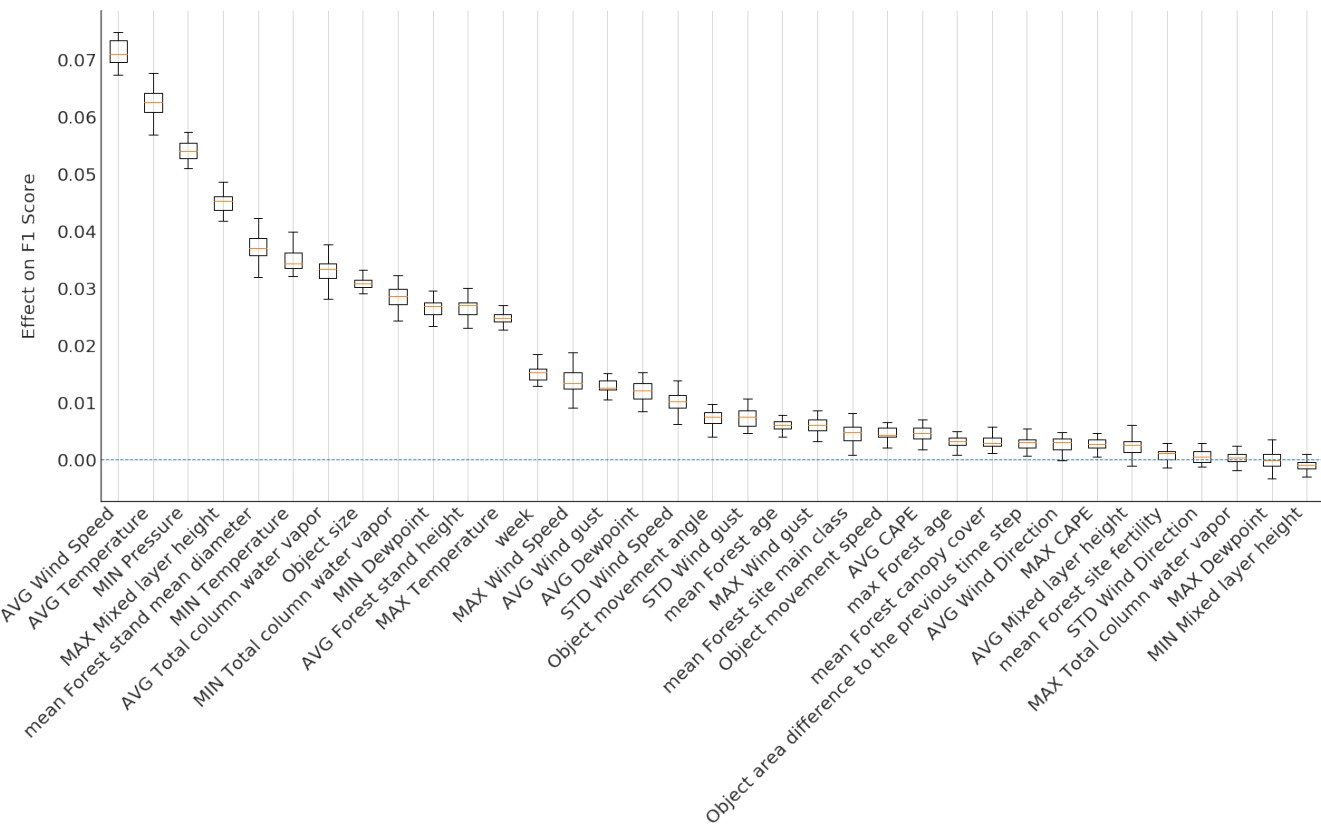

**Figure 7.** Permutation feature importance using the GP classification method trained with the randomly selected national dataset. The higher effect on the F1 score is (y-axis), the bigger is the significance.

example cases, the model performs well (storms Tapani and Pauliina) and in one (storm Rauli) with less accurate prediction results.

### 4.2.1  Event 1: Extratropical Storm Tapani (26 December 2011)

The first example is one of the most known extratropical storms in Finland. Storm Tapani, also known as Cyclone Dagmar (Kufeoglu and Lehtonen, 2015), was a rare winter storm, causing broad and long-lasting electricity interruptions. Extreme wind gusts of over $30 \text{ m s}^{-1}$ caused widespread damage, especially in the southern and western parts of the country. Approximately 570 000 households were left without electricity, causing 30 million euros repair costs and 80 million euros of monetary compensation for electricity distribution companies to their customers (Hanninen and Naukkarinen, 2012). Exceptionally warm December and the Boxing day being the warmest in 50 years (Finnish Meteorological Institute, 2011) resulted in wet and unfrozen soil. Thus, the trees were poorly anchored and exposed to significant storm damage.

Figure 8a represents the outage prediction (raster-covered areas) and the actual, true classes (numbers) based on the damage data at 15:00 UTC, 26 December 2011. Wide areas in central and western parts of Finland are predicted to have *high* (class 2) damages. The predicted class is in line with the true class. Also, the damage areas of the storm correlate with the wind gust observations of the Finnish Meteorological Institute. The strongest gusts occurred in western (15-27 $\mathrm{m\,s^{-1}}$) and southern (18-28 $\mathrm{m\,s^{-1}}$) Finland and north-western part of Lapland (13-31$\mathrm{m\,s^{-1}}$) (Finnish Meteorological Institute, 2020). In the rest of Finland, the maximum wind gusts remained between 10-15 $\mathrm{m\,s^{-1}}$, and therefore the damages were minor. Overall, the model predicted the damages accurately in this particular example.

### 4.2.2 Event 2: Extratropical Storm Rauli (27 August 2016)

Extratropical storm Rauli was an exceptionally strong summer storm, especially regarding the impacts. It caused severe damages to the power grid in the western and middle parts of Finland for various reasons. The trees were carrying leaves, the soil was wet after a rainy August, the strong wind areas of Rauli were widely spread, and the solar radiation was intensifying the wind gusts during the afternoon (Finnish Meteorological Institute, 2016). Rauli was impacting especially the middle and southern parts of Finland, which are also the most densely populated areas. The power outages were increasing rapidly in the middle part of Finland, starting at midday and reaching the highest values, 200 000 households without electricity (Ilta-Sanomat, 2016), around 5 pm. The winds were blowing exceptionally long, nearly 24 hours. The typical duration of summer storms is between 6-12 hours.

Figure 8b shows the predicted outages and true classes at 12:00 UTC, 27 August 2016. In this particular time step, the model is over-predicting the class, however, the predicted outage area seems to correlate with the wind gust maximums of that afternoon. The strongest wind gusts were measured in the southern and middle parts of the country, maximum gusts reaching on land stations up to 24,9 $\mathrm{m\,s^{-1}}$ (Klemettilä, Vaasa and Maaninka, Pohjois-Savo) and on wide areas up to 20 $\mathrm{m\,s^{-1}}$ apart from the northern part of Finland.

### 4.2.3 Event 3: Extratropical storm Pauliina (22 June 2018)

The last example is a strong extratropical storm, called Pauliina (Finnish Meteorological Institute, 2018) that caused numerous power outages in Finland. The most significant part of the power outages happened in the network of power grid company JSE included in the local dataset. The highest peak in the damages was reached between 6 and 8 pm with over 28 000 households without electricity. The strongest wind gust on land reached 22,7 $\mathrm{m\,s^{-1}}$ in Helsinki, Kumpula observation station, and the inland gusts were widely between 15-20 $\mathrm{m\,s^{-1}}$ (Finnish Meteorological Institute, 2020; Finnish Meteorological Institute, Twitter). The strong wind gusts continued until the dawn of the 23rd of June.

Figure 8c presents the predicted and true damage classes at 01:00, UTC, 22 June 2018. We chose extratropical storm Pauliina as an example storm for two reasons: 1) Pauliina represents a *low damage* class 2) Pauliina represents a rare, summer-season extratropical storm. Figure 8c shows the predicted and true classes correlating. While weather warnings were issued to large areas in southern and middle parts of Finland, (myrskyvaroitus.com, 2018) predicted and true damage to the power grid occurred in a relatively small geographical area.

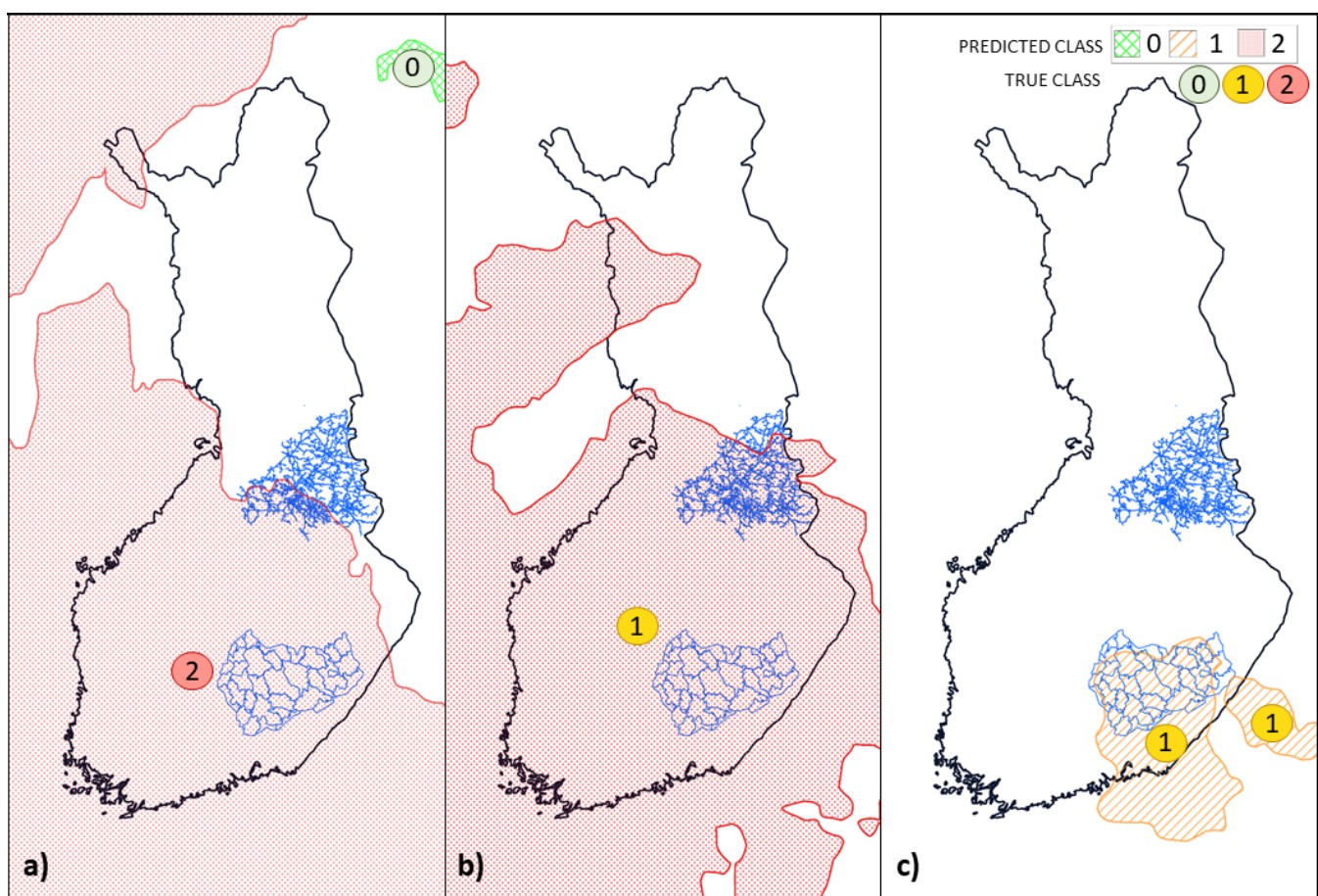

**Figure 8.** Selected examples a) Extratropical storm Tapani (26 December 2011 11:00) b) Extratropical storm Rauli (27 August 2016 10:00) c) Extratropical storm Pauliina (22 June 2018 01:00), produced by employing the SVC model trained with the national dataset. The storm objects are colored based on the predicted class while the true class is stated as a colored number over the object. The time is represented as UTC time.

## 5 Discussion and conclusions

This paper introduces a novel method to predict the damage potential of extratropical storms to power grids. The method consists of identifying storm objects by contouring surface wind gust fields with the 15 m s$^{-1}$ threshold along with pressure objects with a 1000 hPa threshold, tracking the objects, and then classifying them into three classes based on their damage potential to the power grid. For the classification task, we evaluated five different machine learning methods, all employing in a total of 35 predictive features and trained with eight years of power outage data from Finland.

Both Gaussian Processes and Support Vector Classifiers provided good results. The model recognizes harmful storm objects well and can distinguish extremely harmful objects among others adequately. While the results still leave a lot to improve, the

developed model can be already used to support decisions in power grid companies. In some cases, the model is able to provide a more specific and geospatially accurate prediction of potential damage to the power grid than, for example, weather warning. The evaluation was, however, based on the ERA5 reanalysis data. Using the method in an operational setting would require weather prediction data, which introduces additional uncertainty to the outage prediction.

The presented object-based approach has both advantages and disadvantages. Extracting storm objects in advance prepro­cesses the data for machine-learning techniques, such as RFC, which do not perform feature learning. It enables machine-learning methods to focus only on the relevant parts of the data. Methods not containing feature learning, such as RFC and logistic regression, have been found to outperform neural networks for forest (Hart et al., 2019) and weather data (Tervo et al., 2019). It also leads to significantly faster training times. Processing objects instead of the grid makes it also easier to track and use object attributes such as age, speed, and movement. Moreover, objects are easy to visualize, and user interfaces may be enriched with related actions such as tracking and alarms.

On the other hand, storm objects use only aggregated attributes, which may decrease the classification accuracy when predictive features vary significantly under the storm object area. Several machine-learning methods, i.e., deep neural networks, could be trained to employ those local features to gain better accuracy. Such methods could also utilize three-dimensional data.

The fixed threshold of wind gust and pressure were used to extract the storm objects in this paper. Although the previous studies indicate the critical threshold of wind gust speed to be the same for the almost entire geospatial domain of this work (Gardiner et al., 2013), it would be beneficial to adapt the threshold based on the geographic location using, for example, storm severity index (SSI) originally introduced in Leckebusch et al. (2008). Moreover, the correct threshold may vary depending on the data source.

The work opens several possible avenues for further studies. It would be interesting to compare the current solution with a grid-based approach and deep neural networks. Including data on soil moisture, soil temperature, and leaf index would most likely enhance the results, if available with sufficient spatial and temporal resolution, since they would provide critical information about the environmental conditions. Different thresholds could be investigated as well, especially for pressure objects where lower thresholds might yield better results. By design, applying the method to other regions is possible, but it is subject to the availability of power outage records, forest inventory, impact and meteorological data. For the classification task, carefully designed Bayesian networks could provide good results as well. Especially in the randomly selected test set, data may be autocorrelated, which may lead to unrealistically good results. We have addressed this issue by also using a continuous time series (from 2010 to 2011) for the test set. The evaluation could also be extended with a leave-one-day-out or leave-one-week-out method where for each week one day or for each month one week is hold out for validation purposes.

End users, especially expert users like duty forecasters, might benefit from the uncertainty information originating as the probabilistic prediction of the classification model. However, the presentation of such information should be very carefully chosen not to mislead non-expert users for overconfidence.

Experiments in this study were conducted with ERA5 reanalysis and additional forest data. As the method employs common features existing also in various other datasets, data provided by other vendors could be used as well. By employing weather

forecasts as input, this method could be used as a base for a decision support tool and as a part of an existing early warning system for both duty forecasters of national hydro-meteorological centers and operators of electricity transmission companies.

*Code and data availability.* The source code is available in the repositories https://github.com/fmidev/sasse-era5-smartmet-grid and https://github.com/fmidev/sasse-polygon-process. ERA5 data may be downloaded from the Copernicus Climate Data Store: https://cds.climate.copernicus.eu. Forest inventory may be downloaded from LUKE open data service: http://kartta.luke.fi/index-en.html. The power outage data is propriety data which the authors have no property rights to distribute.

## Appendix A: Gaussian distribution fitted to the storm object features

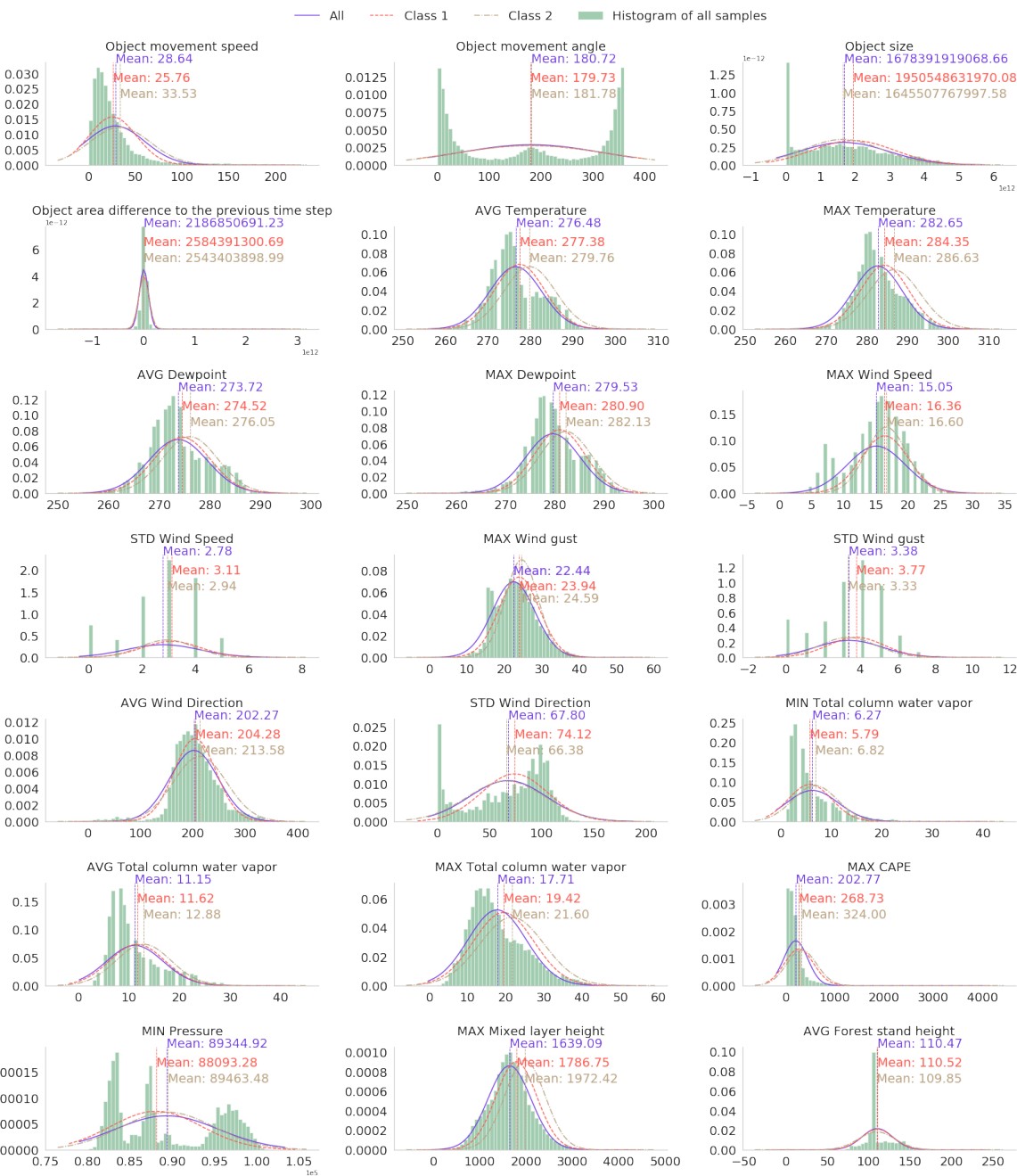

**Figure A1.** Histogram of and fitted Gaussian distribution of selected predictive parameters in **the local dataset**. The Gaussian distribution is fitted separately to all samples and samples with little outages and many outages (classes 1 and 2 specified in Section 3.3).

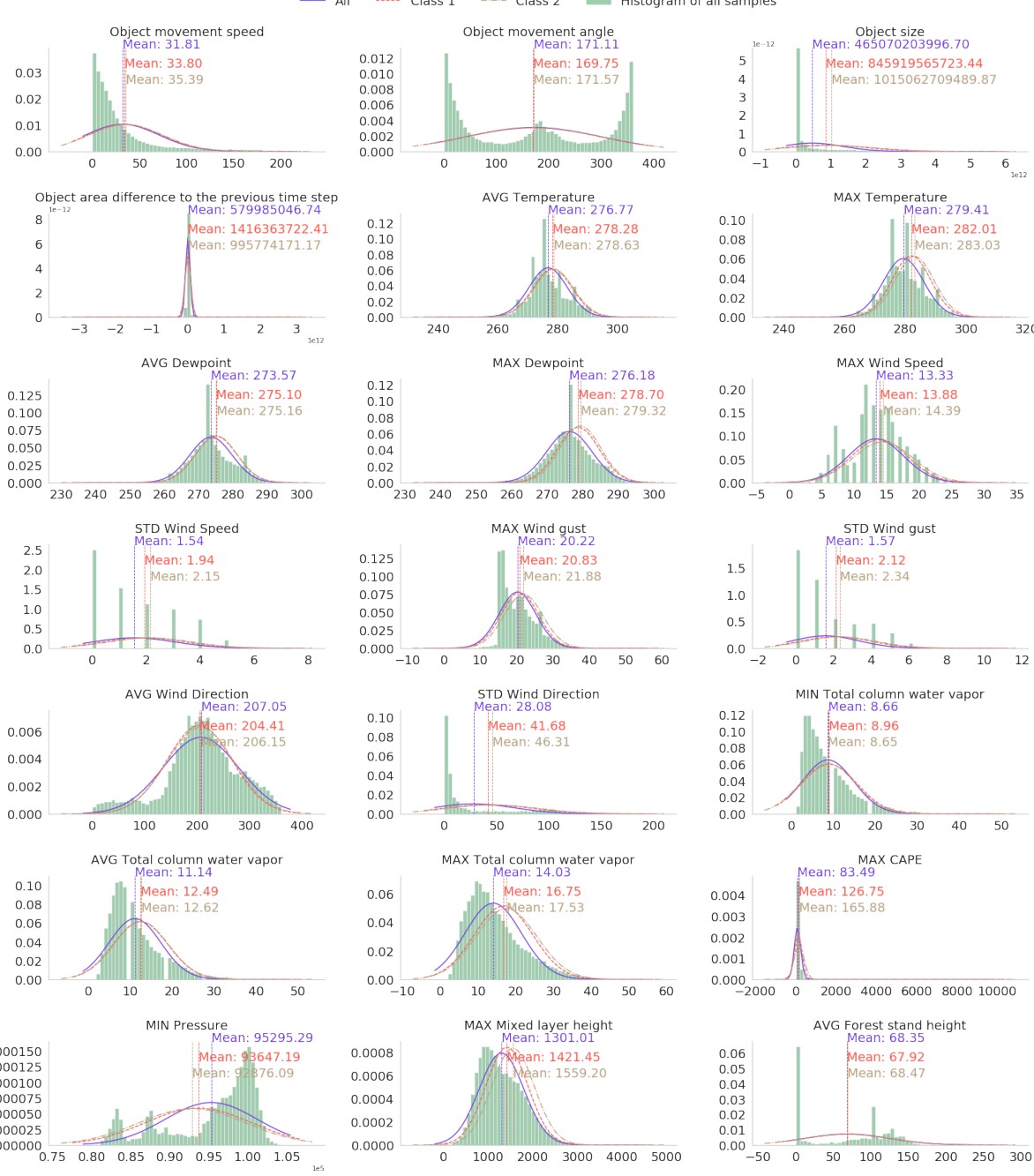

**Figure A2.** Histogram of and fitted Gaussian distribution of selected predictive parameters in **the national dataset**. The Gaussian distribution is fitted separately to all samples and samples with little outages and many outages (classes 1 and 2 specified in Section 3.3).

*Author contributions.* RT conceptualized, designed, and developed the method. IL contributed with meteorological expertise, such as selecting used data and meteorological features along with correct thresholds. IL also helped in analyzing the performance. AM provided supervision from a meteorological perspective and AJ from a machine learning perspective. All contributed in presenting the results.

*Competing interests.* The authors declare that they have no conflict of interests.

*Acknowledgements.* The authors express their gratitude to Järvi-Suomen Energia, Loiste Sähköverkko, and Imatran Seudun Sähkönsiirto for
sharing data and their experience.

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
