# Peer review of "Predicting power outages caused by extratropical storms"

_Natural Hazards and Earth System Sciences, 2020_

## Referee Comment (RC1) · Anonymous Referee #1 · 7 Sep 2020

General remarks

The article investigates windstorm impacts on the power grid in Finland. The authors present a methodology to identify storm objects as polygons and combine them with meteorological and non-meteorological data to predict power outages. They use ERA5 reanalysis data, a national forest inventory and a dataset with information about time and location of power outages in Finland. Storm objects are identified using a fixed wind speed threshold of 15 m/s are tracked in time and space. A large set of meteo­rological and non-meteorological parameters is gathered for each storm object. From these parameters the most relevant are selected and five different methods are used to classify the storm objects with respect to the damage they caused to the power grid using three damage classes. It is tested how well the different methods are able to

predict the class of a storm object using cross-validation. Finally, the best performing classification method is applied to three test cases of severe storms.

In general, the article addresses the very interesting and relevant topic of predicting the impacts of extreme weather events. The authors use state-of-the-art data and methodology. However, there are some issues in the manuscript and there are some parts that need more detailed explanation and discussion. These issues should be addressed before the manuscript is accepted.

The authors use sophisticated methods for classification of storm objects with a large set of parameters. What is missing in the study is an analysis of the relevance of the individual parameters for the classification task. It remains unclear which of the parameters play an important role. It might be, for example, that it is mainly the size of the storm object or the number of transformers under the object that is relevant for the damage, while the standard deviation of wind direction plays a minor role. It would be beneficial to include an analysis of the importance of the parameters, at least for the best performing method, to add more scientific insight to the rather technical aspects of classification task.

The authors should discuss what is the benefit of using storm objects, rather than directly relating wind speeds and other parameters to power outages in a certain area, for example in a grid-based approach. Following the approach in the manuscript, one is able to assign a damage class to the whole area of the storm object. However, this does not provide any information about the specific location of the outage. I would suggest to discuss in more detail what could be the use of such a large-scale damage information for an energy provider (see also my specific comment further below).

In many figures the labels are hardly readable.

The manuscript needs to be checked for English language.

Specific remarks:

Page 3, line 81: What is the spatial resolution of the forest inventory?

Page 3, line 84-88: It could be useful to introduce Figure 1 already here in the data section. This would be helpful for the reader to understand the extraction of storm object feature in section 3.2. You should also go into more detail about the spatial accuracy of the local and national data set.

Page 4, line 97: Can the storm polygons have "holes", if within the area of a polygon areas with winds below 15 m/s exist?

page 4, line 103: Here you mention pressure objects for the first time. Are they defined by the 1000 hPa threshold? Please describe in more detail. Also, when you use the word "object" on its own, it is not clear if you refer to a "storm object" or "pressure object". Therefore you should only use "storm object" and "pressure object". Later you also use the term "wind object".

page 5, algorithm 1: What is the "previous pressure object"? Is it previous in time? Or is there another for-loop that cycles through the pressure objects, which is not mentioned in the algorithm? What is "other object"? You mention "object", without specifying if it is a storm or pressure object. Please revise the algorithm, so that it is easy to understand for the reader.

page 5, line 123-128: From your description it is not clear how you selected the relevant parameters. You write about a fitted Gaussian distribution. How do you fit it, to which data and with which purpose? What is class one and two? What is the criterion for selecting the 35 relevant parameters?

page 7, line 130-131: At this point it is not clear how you define the three classes. To make it easier for the reader, I would suggest to spend some words on how the classes are defined here, or to move this part to page 8, line 155, where the classes are actually introduced.

page 7, line 136-138: You write "the local dataset contains 24,542 storm objects".

Would it be more precise to say that "24,542 storm objects are related to outages in the local outage dataset"? It would be very informative to know how many outages are in the dataset in total and how many of them are NOT related to a storm object. Maybe you can add that information here.

page 7, figure 1a: Can you explain why the network topologies look so different in the northern and southern area? In the north it looks like branches that end somewhere, in the south it rather looks like district boundaries. Figure 1c and d: What is shown here in red color? Number of outages per area? Please add a legend. I would recommend to plot the grid topology with a darker color on top of the shading to increase its visibility.

page 8, line 153-154: Please explain in more detail what is shown in figure 4. Does one dot represent the outages and affected customers related to a specific stom object? Is the line a linear regression?

page 10, table 2: The caption say "Classes for local dataset", but shown are also classes for the national dataset.

page 10, line 153-154: Is "model" the correct term here? Isn't it rather "classification algorithm"?

page 11, equations 1, 2, 3: If you use equations, you need to define the individual variables. Also, the equations are not easily understood without further explanation.

page 14, section 4.1: As far as I can see it is not mentioned in the text which classification algorithm was used for the case examples.

page 15, figure 5: The figures should be as self-explanatory as possible. Please explain in the caption what the numbers represent.

page 16, line 305: The term "cell" is usually used for convective thunderstorms, but not for large-scale winter storms. I would suggest to simply use the word "storm".

page 17, line 307: The authors state that "the model is able to provide a more specific

and geospatially accurate prediction of caused damage to the power grid than for example weather warning." I do not think that this statement is true. If I understand the model correctly, it assigns a damage class to the whole area of a storm object. This area can be quite large, as figure 6a and 6b show. Furthermore, the model provides no geospatial information about where inside this area the damages are expected. I suppose that weather warnings are available for Finland at a much higher spatial resolution. Additionally, weather warnings are released in advance of an event. In this manuscript the authors do not take into account forecast uncertainty. Therefore, a comparison to weather warnings difficult.

Figures A1 and A2: The figure labels are hardly readable and the figure caption is not self-explanatory. There are abbreviations used in the figure titles which are not defined. Please spend some more words on what is shown on the figures. Can you explain the peak at -1000 in the figure titled "speed_self" and "angle_self"? It appears to be completely detached from the rest of the distribution. Why is there no blue line in the figures titled "AVG Wind gust"?

Technical comments

page 2, line 50 "showed that" instead of "showed at"

page 3, lines 63-66: Please check the description of the paper organization. There are missing words and incomplete sentences.

page 7, line 136: Do not use blank spaces to separate numbers in order to prevent line breaks.

---

## Referee Comment (RC2) · Tim Kruschke (Referee) · 30 Sep 2020

The manuscript "Predicting power outages caused by extratropical storms" by Tervo et al. presents a novel method to predict the danger of extra-tropical storms to cause power outages over Finland, which is mainly due to windthrow in forest landscapes. Based on meteorological data taken from the ERA5-reanalysis as well as forest inventory data and power outage information from two local power network companies and the national responsible authority, they developed and tested classification schemes potentially suitable for warning purposes by distinguishing between severe damage events, small damage events, and no damage events. This is certainly a very interesting an relevant topic and deserves publication in NHESS. However, I consider a number of modifications necessary before publishing.

General comments:

a) A general shortcoming I notice in the prediction and its evaluation is the lack of any geographical assignment. In principal the predicted event is just "severe damage", "small damage", or "no damage" for Finland as a whole, just complemented by the polygon(s) of the storm objects. From a user-perspective (electric power network providers etc.) the question is if such a prediction is really useful facing the potential consequences, that is the alert of manpower to fix potential damages to power lines which will be rather concentrated in specific regions for most events. Of course it is better than nothing but I am sure that the method could be easily advanced to provide more regionalized information. The least thing that could have been done is to provide information on the detail level of the (power network) input data. This would mean something like "severe damage in local network 1, small damage in local network 2 and region 3 of the national network".

b) I consider the explanations of the tested classification algorithms as too short. Maybe these different methods are self-explaining for members familiar with a variaty of sophisticated classification schemes and machine-learning but I think for the majority of the NHESS-readership which I assume to be with geoscientific background these methods are hard to assess. I would like the authors to provide a little more information about the general functionality, pros & cons, and existing studies in the context of weather and climate having made use of these approaches. For some approaches like the SVC or the GP, some of this information are already given, for others this is hardly the case.

c) Especially for readers with a geoscientific background (as said, probably the majority of NHESS-readership) it would be interesting to read something about the relative importance of the various factors listed in Tab. 1. I understand that this may be quite different for the different classification schemes. But at least for those schemes eventually assessed to yield the best performance a qualitative summary could be listed, may mentioning the five most important factors in order of relevance.

d) As far as I understand, the evaluation metrics in equations (4)-(7) are standard metrics used in the field of machine-learning based classification. However, what I am missing in these scores is any consideration of the distance between predicted and observed class. Clearly a prediction of "severe damage" in cases of no observed damage and vice versa is worse than predicting "small damage" in these cases. But this is not reflected in any penalty for the given scores. Maybe this is a wise solution given that the classes are very different in population. Otherwise an "algorithm" always predicting no damage might be superior with respect to a score taking this distance into account. I would ask the authors at least to comment on this matter and explain why they do not penalyze larger distance between prediction and observation.

e) I wonder why the authors decided to provide deterministic category predictions. This is to some degree a philosophical discussion but given the nature of the task to make a prediction and further supported by i) the rather arbitrary distinction between event classes and ii) the large number of influential factors (some of them considered in the categorization schemes but many more existing in the real world), I wonder why the authors didn't design a scheme that provides probabilities for the distinct event categories. It is often argued that end-users prefer deterministic predictions but it is clearly a fact that predictions such as produced in this study are subject to significant uncertainty. So, why not making this uncertainty transparent by providing related estimates in the form of probabilities? I do not ask the authors to re-design their whole approach but please comment on this issue. Maybe it is worth considering this as a future extension or advancement of the presented approach.

f) A very general issue is that the authors use the term prediction (and so do I in this review) but in fact the presented approach is based on atmospheric REanalysis data, i.e. it relies on data retroactively produced from observations. I would ask the authors to rephrase respective introductory and conclusive remarks in a way that it becomes clear that this study serves as a general introduction of this approach and a proof of concept while a quasi-operational implementation at weather services or

power network providers would have to be based on actual weather predictions which will introduce additional uncertainty to the final product.

g) Some of the figures need optimization. Please see my respective specific comments.

h) I am not a native English speaker myself, so I usually refrain from judging the language used in manuscripts written by others. However, in this example I have the strong impression that the language should be revised. A particular example are frequently missing definite and indefinite articles ("a" and "the"). Other examples can be found in my specific comments.

Specific comments:

1) line 14: Please revise your citation. This is certainly no person with family name "Re" who is cited here but a institutional citation referring to a publication by the Munich Re.

2) lines 20-21: "...up to 69% compared to previous years". What is meant here? Is it an increase of 69% compared to previous years or a total of 69% of outages in 2011/2013v which are associated with windstorms. If the latter is the case, then please delete "compared to previous years".

3) line 27: "Ulbrich et al. (2009)", not "Ulbrich et al. (Ulbrich et al., 2009)"

4) lines 31-33: Please rephrase to make clear that this sentence contains references to studies contradicting the fore-mentioned studies and their results.

5) line 46: Delete "large-scale storms" and "small-scale storms" and just name the meteorological phenomena themselves as they are now listed in brackets. It is misleading to call hurricanes large-scale if then coming to the extra-tropical storms which are even larger in spatial scale.

6) lines 52-55: The purpose of this sentence is a little unclear to me, especially the

reference to the IPCC-SREX-report. It's fine citing this report but not as one of several/many examples supporting this statement. It is basically the probably most comprehensive summary/review of studies indicating this.

7) line 64: Maybe replace "features" by "storm object features" or "storm object characteristics"

8) lines 68-73: Please indicate the purpose of each dataset in this study, e.g. "the ERA5 atmospheric reanalysis (Hersbach et al., 2019) provides the primary meteorological input data for this study...".

9) lines 74-80: Please indicate explicitely which level you use regarding the ERA5 wind data. I guess it is the 10m-winds but this is not said here. Additionally, you may comment on the issue regarding ERA5 surface winds which is described at https://confluence.ecmwf.int/display/CKB/ERA5%3A+large+10m+winds . As far as I can see this does not affect this study as all problematic occasions of unrealistic high wind speed happened at geographical locations far off the study domain. Still I consider this worth mentioning as some readers may not be aware of this issue in general (so the authors could contribute to a more widespread awareness of this problem) and others may be aware of the problem but not its location and related irrelevance for this study.

10) lines 84-88: What is the specific benefit of using the two local datasets on top of the national dataset for this study? Please comment.

11) lines 97-99: The reason given for using a threshold of 15m/s is valid as long as observed winds are considered. However, ERA5-winds are not observed winds, especially regarding gusts. It's basically model results. It should be noted here already, that at least a little bit of sensitivity tests have been performed yielding 15m/s to be the "best" choice. However, the motivation behind this study to develop a scheme which is applicable for quasi-operational forecasts would imply a transfer to a different source of meteorological data, basically weather predictions. Weather prediction models feature

quite different distributions of surface wind speeds. Hence, for such an application a thorough test of the use of this threshold would be necessary. I would like to point out that there are approaches existing in the published literature on wind damage that make use of thresholds which are tied to the specific wind climatology of respective datasets, e.g. by making use of specific quantiles rather than absolute values.

12) line 103: Do you mean "...connected to objects in preceding timesteps"?

13) line 103: Why do you call this "Algorithm 1" if there is no "Algorithm 2"? Why not simply calling it "Storm tracking algorithm"?

14) line 103-104: Maybe I missed something but it seems to me you are not providing any information about the criterion to define/identify a "pressure object".

15) line 103-104: You mention "the threshold" but such a threshold has not been introduced yet. This is done a few lines below. Please rephrase.

16) line 111: "That means that wind objects are not assumed to move..."

17) line 111: "45km" instead of "45km/h"; and please add "from one hourly timestep to another" to the end of this sentence.

18) line 115: "The first group is a number of object characteristics ... which are calculated ..."

19) line 117-118: Please provide more details how you aggregate. Are the minimum/maximum/average values calculated over all grid boxes identified to belong to the storm object (i.e. exceeding 15m/s)?

20) line 118: Replace "over" by "on"

21) line 119: Replace "features" by "characteristics"

22) line 120: Replace "in the damages" by "to the damages", "support" by "complement", and "with weather parameters" by "for weather parameters"

[Figure]

23) line 121: Replace "aggregated from" by "aggregated over".

24) line 124: Here you mention the samples for class 1 and 2 but the class definition has not yet been introduced. This happens in the next section. Please refer to this section and include a very brief definition of the two classes in this sentence, e.g. "severe damage" and "small damage"

25) line 131: Now you introduce the general class definition (no damage, low damage, high damage) but again the exact definition is found at the very end of section 3.3. Additionally, the thresholds used to distinguish between the classes, especially between the two classes containing damage, seems to be completely arbitrary. AT least there is no reason given why the respective number of outages is considered to be low-damage or high damage.

26) Fig. 1: Looking at the red lines in Fig. 1a & b I get the impression that only the lines for the northern local dataset illustrate actual power lines. The lines for the southern local dataset rather seem to be boundaries of sub-regions or so just as Fig. 1b contains region boundaries. I suggest to use different colors for different types of information. The spatial distribution of outages in Fig. 1c & d seems to having been smoothed. If so, please indicate this and the reason for doing so.

27) lines 143-149 (and especially when reading lines 145-146): The reader immediately wonders why the authors stay with the 15m/s-threshold and why this is not analyzed in terms of quantitative measures. A simple example might be hit rates and false alarm rates or so. It is only in Sec. 4 (lines 248-250) that the authors write that storm identification with 15m/s yields a bet\ ter basis for the following classification. Please refer to this later explanation here.

28) lines 155-158: Eventually the class definitions seem to be set arbitrarily. If there is a reason behind the particular thresholds, please name these.

29) lines 160-161: Why is centering and normalization necessary? Probably for some

classification algorithms but not for all of them, right?

30) lines 162-163: Please describe briefly what the application of SMOTE means and why this is beneficial/necessary.

31) lines 204-206: Why did you choose this specific topology? Did you test others? How is the sensitivity of the results to the networks topology?

32) line 236: Please explain the content of the confusion matrices briefly. Again this is probably clear to people profound in machine-learning based classification but not necessarily to the general readership in geosciences. If I understand correctly, it is simply the ratio of cases for each observed class that is show in the cells for the respective predicted classes, right?

33) Section 4.1: This whole section is where my major comment a) becomes visible. If I understand correctly, it is just the event as a whole which is assigned with the respective category, complemented by the polygon of the storm object(s). Is it possible that different objects of one specific event are assigned with different classes? Fig. 6a seems as if this is possible. On the other hand the northeastern object is outside of Finland, so it is clear that there is no damage (to Finnish power lines) observed. In this context it becomes also visible that intra-object refinement of the classification would be desirable. It makes hardly any sense for a prediction of potential damage to power lines (due to windthrow in forests) that the storm objects extend over the Baltic Sea. I understand that this is due to the primary identification being solely based on the exceedence of the wind speed threshold. However, I ask the authors to thin and comment on my general comment a). Additionally, this case study validation refers to observed wind gusts when qualitatively assessing the credibility of these specific predictions. But the authors made it very clear that the potential damage due to a windstorm depends on many more factors, partly non-meteorological but related to the forests themselves. This raises again the question of relative importance of the various factors.

34) lines 306-307: This sentence ignores the fact that the actual study was based on reanalysis data. Using actual weather predictions - which would be necessary for this prospect mentioned here - would introduce additional uncertainty and very likely lead to worse results than derived in the current study. This does not lower the value of the current study but it is worth mentioning when writing about such potential quasi-operational applicability.

35) line 309: Start the sentence with "Including data on..."

36) line 309: I agree that including data about forest soil and leaf index would probably be beneficial but it is questionable if such data is available in sufficient spatial and temporal resolution and coverage.

37) Appendix A: All text elements and axis labels in figures A1 and A2 are hardly readable.

---

## Short Comment (SC1) · 5 Oct 2020

A thoroughly interesting paper. The methodology for identifying storm is especially interesting. However, there may be a few ways to improve the work presented. More specifically:

1) In lines 46 to 48, the authors claim that modeling power outages caused by extratropical events is an understudied problem. However there are actually several papers that describe a power outage prediction system designed specifically for modeling power outages from extratropical storms that are not cited: Yang et al, https://www.mdpi.com/2071-1050/12/4/1525; and Cerrai et al, https://ieeexplore.ieee.org/abstract/document/8656482

2) In figure 4b, it's unclear why the data contains prominent examples where there are very few or no outages, but have a large number of customers affected. Is this trend real, or is it an artifact of noise in the data?

3) By using week as a predictor variable the authors may be over-fitting. For example, to my knowledge, there's no specific mechanism of why a storm on the 42nd week of the year would be particularly strong. But if you had several examples of strong storms on that week, the model would learn that trend and begin to predict strong outages just because of the week, independent of the actual meteorological characteristics of the storms. There are probably other, less problematic ways to describe seasonal aspects of storms to the model.

4) I would recommend a more rigorous and comprehensive method for validating the model. As discussed in the paper, the k-fold cross-validation approach may not sufficiently isolate temporally or spatially correlated information from the model, and thus inflate the model's performance. The 2010 to 2011 holdout approach is presented as alternative to this approach, but the types of storm events that occur often vary widely from year to year. A leave-one-day/week/month/year-out cross validation (where for each day, week, month, or year in the database you hold out that data, train the model on the remaining data, and predict on the withheld data. Then evaluate the model on all of those results) would provide more compelling results.

---

## Author Comment (AC1) · 8 Oct 2020

**Response to the Comments RC1 on Manuscript Predicting power outages caused by extratropical storms**

Corresponding author: Roope Tervo, roope.tervo@fmi.fi
Oct 7th, 2020

We thank the reviewer for taking the time to read our paper and for giving us insightful, constructive, and extremely valuable comments and improvement suggestions. We have addressed all the comments as accurately and precisely as possible and made the improvements in the manuscript.

In the following, we respond to the comments item-by-item. The referee's comments are indented and with italic typesetting. The authors' comments are with normal typesetting. Direct quotes from the manuscripts are marked with double-quotes.

**Responds to the general remarks**

*General remarks*

*The article investigates windstorm impacts on the power grid in Finland. The authors present a methodology to identify storm objects as polygons and combine them with meteorological and non-meteorological data to predict power outages. They use ERA5 reanalysis data, a national forest inventory and a dataset with information about time and location of power outages in Finland. Storm objects are identified using a fixed wind speed threshold of 15 m/s are tracked in time and space. A large set of meteorological and non-meteorological parameters is gathered for each storm object. From these parameters the most relevant are selected and five different methods are used to classify the storm objects with respect to the damage they caused to the power grid using three damage classes. It is tested how well the different methods are able to predict the class of a storm object using cross-validation. Finally, the best performing classification method is applied to three test cases of severe storms.*

*In general, the article addresses the very interesting and relevant topic of predicting the impacts of extreme weather events. The authors use state-of-the-art data and methodology. However, there are some issues in the manuscript and there are some parts that need more detailed explanation and discussion. These issues should be addressed before the manuscript is accepted.*

*The authors use sophisticated methods for classification of storm objects with a large set of parameters. What is missing in the study is **an analysis of the relevance of the individual parameters for the classification task.** It remains unclear **which of the parameters play an important role**. It might be, for example, that it is mainly the size of the storm object or the number of transformers under the object that is relevant for the damage, while the standard deviation of wind direction plays a minor role. It would be beneficial **to include an***

*analysis of the importance of the parameters, at least for the best performing method, to add more scientific insight to the rather technical aspects of classification task.*

We conducted a permutation feature importance analysis using the Gaussian processes (GP) model and a randomly selected test set of the national dataset. The same model and data are used to produce the case examples.

The manuscript is appended with the following chapters (page 17 in the updated manuscript):

"The relevance of the individual predictive features can be explored by using the permutation test, as done by Breiman (2001). First, the baseline score of the fitted model is calculated using the test set. Then each feature is randomly permuted, and the difference in the scoring function is calculated. The random permutation is repeated 30 times for each parameter, and the average of the results is used. The procedure offers information on how important the feature, the individual parameter, is to obtain good results. It should be mentioned that highly correlated features may get low importance as other features work as a proxy to the permuted feature. Using completely independent features is not, however, possible in weather data since weather parameters are often dependent on each other, and eliminating even the most apparent pairs from used features impaired the results in our experiments.

We used the macro average of F1 defined in Equation 7 as a scoring function and randomly selected test set from the national data. The relevance is shown in Figure 7. Most features show at least little relevance for the results. The first twelve features are more significant than the rest. The most important features contain at least one representative of all meteorological parameters used in training. In other words, all employed meteorological parameters are important for the prediction, while different aggregations are contributing to the "fine-tuning" of the model.

As Figure 7 shows, the most significant parameter regarding our model performance is the average wind speed. Numerous studies support our result of wind being the most important damaging factor (Virot et al., 2016; Valta et al., 2019; Jokinen et al., 2015) that are, however rather highlighting the importance of maximum wind gusts. Surprisingly, in our analysis, the wind gust speed does not belong to the most critical parameters. Instead, maximum mixed layer height, related to the wind gustiness, contributes crucially to the model performance. The dependencies between predictive features might be one reason for some parameters to have lower rank in the results.

The stand mean diameter and height are the most important features regarding the forest parameters, which corresponds to our expectations. Previous studies also state these features to influence the wind damage in forests (Pellikka and Järvenpää,2003) and hence indirectly electricity grids. As Pellikka and Järvenpää (2003) and Suvanto et al. (2016) discuss, also the age of the forest has an impact on

storm damages. However, in the feature importance test, forest age does not seem to contribute significantly to the prediction outcome.

The most important object feature is the size of the object. Object movement speed and direction did not contribute to the results much. However, previous studies indicate that besides the size of the impacted area, the duration of strong winds – i.e., the movement speed of the system – influences also the amount of damage (Lamb and Knud, 1991)."

[Figure]

Figure7. Permutation feature importances using GP classification method trained with the randomly selected national dataset. The higher effect on the F1 score is (y-axis), the bigger is the significance.

*The authors should discuss what is **the benefit of using storm objects, rather than directly relating wind speeds and other parameters to power outages in a certain area, for example in a grid-based approach.** Following the approach in the manuscript, one is able to assign a damage class to the whole area of the storm object. **However, this does not provide any information about the specific location of the outage. I would suggest to discuss in more detail what could be the use of such large-scale damage information for an energy provider (see also my specific comment further below).***

Using storm-objects instead of fitting the models with gridded data is a fundamental design choice of the work. Its benefits and downsides will definitely be an interesting subject to cover. We added the following discussion into the manuscript (page 22, 415-430 in the updated manuscript):

"The presented object-based approach has both advantages and disadvantages. Extracting storm objects in advance, preprocesses the data for machine-learning techniques, such as RFC, which do not perform feature learning. It enables machine-learning methods to focus only on the relevant parts of the data. Methods

not containing feature learning, such as RFC and logistic regression, have been found to outperform neural networks for forest (Hart et al., 2019) and weather data (Tervo et al., 2019). It also leads to significantly faster training times. Processing objects instead of the grid makes it also easier to track and use object attributes such as age, speed, and movement. Moreover, objects are easy to visualize, and user interfaces may be enriched with related actions such as tracking and alarms.

On the other hand, storm objects use only aggregated attributes, which may decrease the classification accuracy when predictive features vary significantly under the storm object area. Several machine-learning methods, i.e., deep neural networks, could be trained to employ those local features to gain better accuracy. Such methods could also utilize three-dimensional data.

Extracting storm objects requires a fixed threshold of wind gust and pressure, which may vary depending on the characteristics of geospatial locations. Nevertheless, the previous studies indicate the critical threshold for wind gust speed to be the same for the almost whole geospatial domain of this work (Gardiner et al., 2013). Moreover, the correct threshold may vary depending on the data source. When extending the geospatial domain or changing the data source, this would become a more serious issue, and different thresholds might be needed. One possibility to determine the optimal threshold might be to use specific quantiles of the parameter values, but this would need further studies."

*In many figures the labels are hardly readable.*

We went carefully through all figures and enlarged the labels.

*The manuscript needs to be checked for English language.*

We carefully checked the language and made corrections to the manuscript.

**Respond to the specific remarks**

*Page 3, line 81: What is the spatial resolution of the forest inventory?*

This information has now been added to the manuscript on page 4, lines 98-101.

"The original geospatial resolution of the data is 16 meters, which is reduced to approximately 1.6 km resolution to speed up the processing. Taking into account the size of extratropical cyclones (diameter ~1000 km) and the wide areas where wind damages typically occur, e.g., near to the cold front, we consider resolution of 1.6 km being sufficiently high for modelling wind storm damages."

*Page 3, line 84-88: It could be useful to introduce Figure 1 already here in the data section. This would be helpful for the reader to understand the extraction of storm object feature in*

*section 3.2. You should also go into more detail about the spatial accuracy of the local and national data set.*

We have improved Figure 1 and moved it in the data section and made it more easily understandable and to have it in a more logical place. Firstly, we separated Figures 1a and 1b from 1c and 1d and improved the figures (pages 5 and page 9 in the updated manuscript). We have also added more information about the structure of the local and national dataset and on the spatial accuracy to the data section (page 4, lines 102-115):

"Power outage data are obtained from two complementary sources. *The national dataset* is acquired from Finnish Energy (Finnish Energy, 2010-2018) who aggregates the data from power distribution companies in Finland. The national data is provided only for research purposes and for areas containing a minimum of six grid companies; this is, for example, to ensure energy users' anonymity. Therefore, the national dataset does not include exact locations of the faults. We have also obtained some parts of the data with better spatial accuracy from two individual power distribution companies. In this paper, we name this data to the local dataset. In the local dataset, the fault locations are reported in relation to transformers, i.e, the spatial resolution of the outages vary between few meters to kilometers.

Figure 1 illustrates the geographical coverage of the power outage data. The local dataset contains all outages from 2010 to 2018 from the northern area (Loiste) and outages related to major storms in the southern area (JSE), shown in Figure 1a. The national dataset contains all outages in Finland from 2010 to 2018 divided into five regions, shown in Figure 1b. The national dataset contains in total 6 140 434 outages with relatively low geographical accuracy. On the other hand, the local dataset represents a substantially smaller geographical area with a good geographical accuracy but contains only 22 028 outages in total. We train our classification models, described in more detail in Chapter 3.4, with both datasets to evaluate their performance for different types of data."

[Figure]

**Figure 1.** (a) Geographical coverage of the outage data (local dataset). The red lines represent the power grid of Loiste (northern grid company) and the green lines the operative areas of JSE (southern grid company). Outages of the local dataset are collected from both of these areas. (b) Regions in the national outage dataset. Outages are gathered from the whole Finland and aggregated to the regions shown in the figure.

*Page 4, line 97: Can the storm polygons have "holes", if within the area of a polygon areas with winds below 15 m/s exist?*

The contouring algorithm is capable of finding interior rings of the polygons. The used wind gust fields did not, however, contain any such cases. Thus one storm object represents a solid area (polygon).

This information has been added to the updated manuscript on page 5, lines 125-126.

*page 4, line 103: Here you mention pressure objects for the first time. Are they defined by the 1000 hPa threshold? Please describe in more detail. Also, when you use the word "object" on its own, it is not clear if you refer to a "storm object" or "pressure object". Therefore you should only use "storm object" and "pressure object". Later you also use the term "wind object".*

We clarified these paragraphs on page 5, lines 124-138, and revised the use of the word "object". In particular, we describe the object identification and tracking method following:

"Storm objects are identified by finding contour lines of wind gust fields using 15 ms⁻¹ thresholds from the ERA5 surface level grid with a time step of 1 hour. The contouring algorithm is capable of finding interior rings of the polygons. The used wind gust fields did not, however, contain any such cases. Thus one storm object represents a solid area (polygon) where hourly maximum wind gust exceeds 15 ms⁻¹ during one particular hour. The threshold of 15 ms⁻¹ is selected as different sources indicate Finland being vulnerable to windstorms and rather moderate winds (from 15

ms−1) causing damages to forests (Valta et al., 2019; Gardiner et al., 2013). Valta et al. (2019) developed a method to estimate the windstorm impacts on forests by combining the recorded forest damages from the nine most intense storms and their observed maximum inland wind gusts. According to the formula developed in the study, the inland wind gusts of 15 ms$^{-1}$ alone result in forest damages of 1800 m$^3$.

We also identify pressure objects by finding contour lines using a 1000 hPa threshold to connect potentially distant wind objects around the low-pressure center to the same storm event.

After identification, storm objects are connected to other storm objects around the common low-pressure objects and to the storm and pressure objects in preceding timesteps using Algorithm 1. Each object having pressure objects or preceding objects within the threshold is assigned to the same storm event and gets the same storm ID. Single storm objects without nearby pressure objects or preceding objects are left without ID as they are not assumed to be part of any storm."

*page5, algorithm1: What is the"previous pressure object"? Is it previous in time? Or is there another for-loop that cycles through the pressure objects, which is not mentioned in the algorithm? What is"other object"? You mention "object", without specifying if it is a storm or pressure object. Please revise the algorithm, so that it is easy to understand for the reader.*

The algorithm description has been updated to be more explicit. The readability may have been affected a little bit, but we believe this is a better and more precise way to describe the tracking algorithm. Meritoriously notified questions about previous objects and object types are addressed as well.

The updated algorithm is listed below and updated to the manuscript.
* * *
**Algorithm 1** Storm tracking
* * *
**Input**

    Wind and pressure objects $S_o$ arranged by time

    *pressure distance threshold*

    *wind distance threshold*

    *speed threshold*

    *time step*

**Output**

    Connected wind and pressure objects with storm $ID$

**for all** wind and pressure object $O_{w|p} \in S_o$ **do**

    *current time* ← time of the object $O_{w|p}$

    *previous time* ← *current time* − *time step*

    Current time pressure objects $S_p^c$ ← pressure objects having centroid within *pressure distance threshold* from

        object $O_{w|p}$ centroid and time stamp *current time*

    Previous time pressure objects $S_p^p$ ← pressure objects having centroid within *speed threshold* from

        object $O_{w|p}$ centroid and time stamp *previous time*

    Current time wind objects $S_w^c$ ← wind objects having centroid within *wind distance threshold* from

        object $O_{w|p}$ centroid and time stamp *current time*

    Previous time wind objects $S_w^p$ ← wind objects having centroid within *speed threshold* from

        object $O_{w|p}$ centroid and time stamp *previous time*

    **if** pressure object $O_p^c \in S_p^c$ exists with $ID$ **then**

        Use pressure object $O_p^c$ $ID$

    **else if** pressure object $O_p^p \in S_p^p$ exists with $ID$ **then**

        Use previous time pressure object $O_p^p$ $ID$

    **else if** wind object $O_w^c \in S_w^c$ exists with $ID$ **then**

        Use wind object $O_w^c$ $ID$

    **else if** wind object $O_w^p \in S_w^p$ exists with $ID$ **then**

        Use previous time wind object $O_w^p$ $ID$

    **else if** wind or pressure object $O_{w|p}^p \in S_w^p \cup S_p^p$ exists without $ID$ **then**

        Give new $ID$ to the previous object $O_{w|p}^p$ and current object $O_{w|p}$

    **else**

        Leave object $O_{w|p}^p$ without $ID$

    **end if**

**end for**
* * *
*page5, line123-128: From your description it is not clear how you selected the relevant parameters. You write about a fitted Gaussian distribution. How do you fit it, to which data and with which purpose? What is class one and two? What is the criterion for selecting the 35 relevant parameters?*

We clarified the description as follows in the updated manuscript on pages 6-8, lines 156-173:

"We selected the 35 parameters based on two main factors: First, we prepared a list of potential parameters detected in related studies e.g. Suvanto et al. (2016); Peltola et al. (1999); Valta et al. (2019), or identified through the empirical experience of duty forecasters (Weather and of Finnish Meteorological Institute Duty forecasters, 05/2020). Second, we selected the relevant parameters, which were available to us or accessible with reasonable effort. However, some possibly important parameters,

like soil temperature from ERA5 reanalysis were left out because of the slow downloading process.

After the preliminary selection of the parameters, we conducted dozens of light experiments using different combinations of parameters and models to find the best possible setup. To this end, we fitted Gaussian distribution to each parameter using at first all samples, then samples with few outages, and finally with many outages (classes 1 and 2 specified in Section 3.3). While many other distributions are known to suit better modelling particular parameters, such as Gamma in precipitation, Weibull in wind speed, and Lognormal in cloud properties (Wilks, 2011), Gaussian distribution is a sufficient simplification to help in selecting relevant parameters. We inspected visually the differences between fitted Gaussian distributions to deduce the potential relevance of the parameter. Supposedly the distribution of one parameter is different for all samples and samples with many outages. In this case, the classification method may exploit the parameter to predict the damage potential of the storm object. Distribution of some selected parameters is shown in Appendix A. In total, 35 parameters, shown as bolded in Table 1, were chosen for the final classification. ”

*page 7, line 130-131: At this point it is not clear how you define the three classes. To make it easier for the reader, I would suggest to spend some words on how the classes are defined here, or to move this part to page 8, line 155, where the classes are actually introduced.*

We restructured the text to introduce classes on page 10, line 202 (originally on page 8, line 155), as you suggested.

*page 7, line 136-138: You write "the local dataset contains 24,542 storm objects". Would it be more precise to say that "24,542 storm objects are related to outages in the local outage dataset"? It would be very informative to know how many outages are in the dataset in total and how many of them are NOT related to a storm object. Maybe you can add that information here.*

Using only storm objects related to outages would result in overestimating predictions as the classification model would not see any "harmless" class 0 samples in the training process and assume every sample to cause damage. Thus, we also consider storm objects which are not related to any outage.

The local dataset contains 24 542 storm objects and 5 837 outages connected to 2 363 storm objects. Thus 22 179 storm objects in the local dataset have not caused any outages. The local power outage data contains 16 191 outages, which can not be connected to any storm object. The national dataset contains 142 873 storm objects and 5 965 324 outages connected to 33 796 storm objects. 109 077 storm objects are not connected to any outages, and 175 110 outages can not be connected to any storm object.

We added this information to the manuscript on page 9, lines 174-179.

*page 7, figure 1a: Can you explain why the network topologies look so different in the northern and southern area? In the north it looks like branches that end some where, in the south it rather looks like district boundaries. Figures 1 c and d: What is shown here in red color? Number of outages per area? Please add a legend. I would recommend to plot the grid topology with a darker color on top of the shading to increase its visibility.*

The differences between the network topologies are simply explained by the data we have received from the two individual companies. From the northern company (Loiste), we received a shapefile of their grid. The southern company (JSE) provided their operational areas instead of the grid topology. Therefore, these two topologies look so different, even though in reality also JSE's grid looks similar compared to Loiste.

We have now separated Figures 1a and 1b from 1c and 1d and improved the figures based on the suggestions (Pages 5 and 9 in the updated manuscript). See also the reply to the second comment about the spatial accuracy of datasets.

[Figure]

**Figure 1.** (a) Geographical coverage of the outage data (local dataset). The red lines represent the power grid of Loiste (northern grid company) district and the green lines the operative areas of JSE (southern grid company). Outages of the local dataset are collected from both of these areas. (b) Regions in the national outage dataset. Outages are gathered from the whole Finland but aggregated to the regions shown in the figure.

[Figure]

**Figure 2.** (a) Spatial distribution of the outages in the JSE network (southern area), data gathered between 2010 and 2018. (b) Spatial distribution of the outages in the Loiste network (northern area), data gathered between 2010 and 2018.

*page8, line153-154: Please explain in more detail what is shown in figure 4. Does one dot represent the outages and affected customers related to a specific storm object? Is the line a linear regression?*

One dot indeed represents the outages, but it may not be related to any specific storm. The line is trendline (linear regression). We also added a legend to the figure and extended the description in the manuscript on page 10, lines 196-201 following:

"Figure 5 renders how many customers are typically affected by one outage. The figure contains all outages in the dataset, whether they are related to a storm or not. In the local dataset, usually, 20-30 customers lose electricity in one outage. In the national dataset, only six customers usually lose electricity in one outage. We assume that this roots to different network topologies in other areas."

The original manuscript also contained an error. The original manuscript stated that 200-300 would be typically affected, which is wrong. One outage usually affects from 6 to 30 outages depending on the dataset. We corrected this.

*page 10, table 2: The caption say "Classes for local dataset", but shown are also classes for the national dataset.*

We compliment, and corrected this on page 12, table 2.

*page 10, line 153-154: Is "model" the correct term here? Isn't it rather "classification algorithm"?*

We assume that this refers to page 10, lines 163-164. The "model" is normally used in this context in machine-learning literature. We see the word "algorithm" to refer more to heuristic algorithms instead of models that are fitted to the data. Another option would also be "method", but it may be confused with the overall method, including storm identification and tracking.

We see that the word "model" is the best term in this context.

*page 11, equations 1, 2, 3: If you use equations, you need to define the individual variables. Also, the equations are not easily understood without further explanation.*

The definitions of the variables are fundamental for equations, and we added them to the manuscript. They should help to understand the equations. We also added references for all kernels used in this work. As the used kernels are widely used standard kernels, we prefer to omit a more detailed explanation to keep the text concise and readable.

*page 14, section 4.1: As far as I can see it is not mentioned in the text which classification algorithm was used for the case examples.*

Gaussian processes (GP) was used in case examples. Thus, we also analyzed feature importances using GP.

This information is added to the manuscript on page 19, line 354. We also changed a conclusion slightly on page 22, lines 409 to form:

"Both Gaussian Processes and Support Vector Classifiers provided good results. [...]"

The original statement in the conclusion honoured only SVC, which is inconsistent with results. SVC and GP provided almost similar performance.

*page 15, figure 5: The figures should be as self-explanatory as possible. Please explain in the caption what the numbers represent.*

We added the following information to the manuscript on page 19:

"Each cell of the confusion matrices represents a share of predictions having a corresponding combination of predicted and true class. For example, the middle right cell tells the share of samples belonging to class 1 but predicted to have class 2."

*page 16, line 305: The term "cell" is usually used for convective thunderstorms, but not for large-scale winter storms. I would suggest to simply use the word "storm".*

Good point, this has been changed to "storm object".

*page 17, line 307: The authors state that "the model is able to provide a more specific and geospatially accurate prediction of caused damage to the power grid than for example weather warning." I do not think that this statement is true. If I understand the model correctly, it assigns a damage class to the whole area of a storm object. This area can be quite large, as figure 6a and 6b show. Furthermore, the model provides no geospatial*

*information about where inside this area the damages are expected. I suppose that weather warnings are available for Finland at a much higher spatial resolution. Additionally, weather warnings are released in advance of an event. In this manuscript the authors do not take into account forecast uncertainty. Therefore, a comparison to weather warnings difficult.*

We acknowledge that the comparison with weather warnings can be challenging. As the referee mentions, the model's ability to provide more specific and geospatially accurate information than weather warnings is not a straightforward issue. We mention the geospatial accuracy because, in some cases, the storm object areas are not as big as in 6a and 6b (8a and b in updated manuscript), which are two examples of extremely strong storms. This was the case, for instance, with the extratropical storm, Pauliina where the yellow level of wind warnings was issued to wide areas in central and southern Finland and orange level of wind warnings to the south (see attached figure). This broad wind warning likely leads to all power companies in southern and central Finland being alert and possibly overpreparing for the event. Another important aspect of this work compared to weather warnings is an analysis of inflicted power outages, which can give an insight to power grid operators and duty forecasters about the impacts of forecasted warnings.

Nevertheless, because of the problematic task to indeed take into account the uncertainty of the forecast, we decided to modify the paragraph and update the manuscript (page 21, line 398-402) with a comment about the forecast uncertainty:

"While weather warnings were issued to large areas in southern and middle parts of Finland (Myrskyvaroitus, 2018), predicted and true damage to the power grid occurred in a relatively small geographical area. This example shows the potential added value of impact estimation for power grid operators. However, in this example, we do not take into account the uncertainty of the weather forecasts before the event. Therefore, it is challenging to compare issued warnings with the model performance purely."

We also added the following clarification to the introduction (page 3, lines 70-74):

""[...] The ERA5 atmospheric reanalysis (European Centre for Medium-Range Weather Forecasts, 2017) provides the primary meteorological input data for this study, while the national forest inventory provided by The Natural Resources Institute Finland (Luke) is used to represent the forest conditions in the prediction. Finally, historically occurred power outages from two sources are used to train the model. However, the operational use of the model would require the use of weather prediction data instead of reanalysis."

And following clarification to the conclusion (page 22, line 412-414):

"The evaluation was, however, based on the ERA5 reanalysis data. Using the method in operations would require weather prediction data, which introduces additional uncertainty to the outage prediction."

[Figure]

*Figures A1 and A2: The figure labels are hardly readable and the figure caption is not self-explanatory. There are abbreviations used in the figure titles which are not defined. Please spend some more words on what is shown on the figures. Can you explain the peak at -1000 in the figure titled "speed_self" and "angle_self"? It appears to be completely detached from the rest of the distribution. Why is there no blue line in the figures titled "AVG Wind gust"?*

We reduced the number of shown parameters to enlarge label size. We also replaced "speed_self", "angle_self", "area_m2", and "area diff" with corresponding feature names listed in Table 1. We added the following caption to the figures so that the figures should be self-explanatory:

"Histogram of and fitted Gaussian distribution of selected predictive parameters in the local dataset. The Gaussian distribution is fitted separately to all samples and samples with little outages and many outages (classes 1 and 2 specified in Section 3.3)."

Peaks at -1000 represent missing values. We dropped samples with missing values, which changed the fitted distributions a little. In particular, the differences between the mean values of the distributions reduce, which makes the deduction a little more challenging. Nevertheless, the same parameters still stand out in the analysis.

Fitted Gaussian distributions marked with the blue line have been missing in the original Figures A1 and A2 because of missing values. After dropping all samples with missing values (technically all rows having values -1000 and np.nan), the fit is successful also to AVG Wind gust, MAX Wind gust, and STD Wind gust, and mean Forest stand mean height.

Figures are updated in the manuscript and shown below.

[Figure]

Figure A1. Histogram of and fitted Gaussian distribution of selected predictive parameters in the local dataset. The Gaussian distribution is fitted separately to all samples and samples with little outages and many outages (classes 1 and 2 specified in Section 3.3).

[Figure]

Figure A2. Histogram of and fitted Gaussian distribution of selected predictive parameters in the national dataset. The Gaussian distribution is fitted separately to all samples and samples with little outages and many outages (classes 1 and 2 specified in Section 3.3).

**Technical comments:**

*page 2, line 50 "showed that" instead of "showed at"*

We did this correction with compliments.

*page 3, lines 63-66: Please check the description of the paper organization. There are missing words and incomplete sentences.*

We changed and modified the paragraph as follows (Page 3, lines 75-79):

"This paper is organized as follows: Chapter 2 presents the used data, which is followed by a step-by-step method description in Chapter 3. Chapter 3.1 discusses identifying storm objects and explains the storm tracking algorithm. Chapter 3.2 considers predictive features containing both storm and forest characteristics. Chapter 3.3 discusses how to define labels of storm objects based on the outage data. Chapter 3.4 describes the used machine learning methods. In Chapter 4, we discuss the performance of the method. Finally, Chapter 5 includes discussion and conclusion."

*page 7, line 136: Do not use blank spaces to separate numbers in order to prevent line breaks.*

We prevented line breaks in the middle of numbers using the latex \mbox command but preferred to keep spaces for clarity.

---

## Author Comment (AC2) · 8 Oct 2020

**Response to the Comments RC2 on Manuscript Predicting power outages caused by extratropical storms**

Corresponding author: Roope Tervo, roope.tervo@fmi.fi
Oct 7th, 2020

We are thankful for the precious and detailed comments and good improvement suggestions. We thank the reviewer for reading our paper carefully. In the following replies, we have addressed the comments as accurate and clear as possible and made the improvements in the manuscript.

The referee's comments are indented and with italic typesetting. The authors' comments are with normal typesetting. Direct quotes from the manuscripts are marked with double-quotes.

**Responds to the general remarks**

*The manuscript "Predicting power outages caused by extratropical storms" by Tervo et al. presents a novel method to predict the danger of extratropical storms to cause power outages over Finland, which is mainly due to windthrow in forest landscapes.Based on meteorological data taken from the ERA5-reanalysis as well as forest inventory data and power outage information from two local power network companies and the national responsible authority, they developed and tested classification schemes potentially suitable for warning purposes by distinguishing between severe damage events, small damage events, and no damage events. This is certainly a very interesting and relevant topic and deserves publication in NHESS. However, I consider a number of modifications necessary before publishing.*

*General comments:*

*a) A general shortcoming I notice in the prediction and its evaluation is the lack of any geographical assignment. In principle the predicted event is just "severe damage", "small damage", or "no damage" for Finland as a whole, just complemented by the polygon(s) of the storm objects. From a user-perspective (electric power network providers etc.) the question is if such a prediction is really useful facing the potential consequences, that is the alert of manpower to fix potential damages to power lines which will be rather concentrated in specific regions for most events. Of course it is better than nothing but I am sure that the method could be easily advanced to provide more regionalized information. The least thing that could have been done is to provide information on the detail level of the (power network) input data. This would mean something like "severe damage in local network 1, small damage in local network 2 and region 3 of the national network".*

> The prediction is done for each polygon separately. Typically, such polygons cover only parts of Finland at the time. Thus, we do not predict the amount of damage to the whole of Finland, but only to the areas affected by an extratropical storm. Therefore, power grid operators could receive information about whether the storm hits the

eastern or western part of the country and whether the damage in this region is expected to be light or severe. Moreover, in cases where a storm consists of several separate polygons, we are able to distinguish the damage potential of each polygon. Some examples of the coverages are illustrated in the manuscript case examples, Figure 8 (also attached below). The two first examples are extreme cases where coverage is exceptionally broad, while the third example represents a more typical geographical scale. We also attached two other examples to this response to illustrate a typical geographical scale of the prediction.

We clarified the geographical area in the introduction of the updated manuscript on page 3, lines 63-66:

"We present a novel method to identify, track, and classify extratropical storm objects based on how much power outages they are expected to induce. We adapt convective storm object detection (Rossi (2015), Tervo et al. (2019), Cintineo et al. (2014)) to find potentially harmful areas from extratropical storms by contouring objects from pressure and wind gust fields. Instead of highly-localized convective storms, we aim at larger but still regional geospatial accuracy so that, for example, damages in western and eastern Finland can be distinguished. [...]"

and in the chapter 3 Method on page 4, lines 117-122:

"We predict power outages by classifying storm objects identified from gridded weather data into three classes based on a number of power outages the storm can typically cause. The overall process contains the following steps: (1) identifying storm objects from weather fields by finding contour lines of some particular threshold, (2) tracking the storm object movement, (3) gathering features of the storm objects, and (4) classifying each storm object individually. The classification is conducted to each storm object separately to distinguish the different damage potential of each object. Tracking is, however, necessary to gather necessary features such as object movement speed and direction. Next, we discuss these phases in more detail."

[Figure]

(a) Tapani, 26 December 2011 11:00    (b) Rauli, 27 August 2016 10:00    (c) Pauliina, 22 June 2018 01:00

**Figure 8.** Selected examples (a) Extratropical storm Tapani (b) Extratropical storm Rauli (c) Extratropical storm Pauliina produced employing SVC model trained with national dataset. The storm objects are colored based on the predicted class while the true class is stated as a colored number over the object. The time is represented as UTC time.

[Figure]

Left: unnamed storm, 11th September 2010 16:00 UTC,
Right: Eino, 17th November 2013 14:00 UTC

Processing polygons instead of grid data simplifies and creates a clear presentation for the end-users. This manuscript presents the potential damage areas (storm objects) on the map, where the end-user can visually inspect whether the object

intersects with the power grid. It is easy to calculate in the operative user interface, for example, how many transformers are affected or even anticipated monetary losses.

A geographical aspect has indeed been omitted from the evaluation of the method. The method may work better in one region than another. However, performing a reasonable and descriptive regional evaluation is a complicated task, and we argue that it would cause more confusion than bring value. Consider, for example, an unnamed storm example on 11th September 2010, in the figure above. The polygon is correctly classified into class 2 as it caused many outages in south-western Finland. The polygon also slightly intersects south-eastern Finland. Should it be included in the eastern Finland metrics? If included, it would cause poor performance in that region since it is a class 2 polygon but still did not cause many outages in eastern Finland. If excluded, the proper ground for excluding should be selected, and the reader should be strictly aware of the ground and its consequences.

Thus, we argue that to be concise and clear, showing aggregated metrics describes the performance better than regional ones.

*b) I consider the explanations of the tested classification algorithms as too short. Maybe these different methods are self-explaining for members familiar with a variety of sophisticated classification schemes and machine-learning but I think for the majority of the NHESS-readership which I assume to be with geoscientific backgrounds these methods are hard to assess. I would like the authors to provide a little more information about the general functionality, pros & cons, and existing studies in the context of weather and climate having made use of these approaches. For some approaches like the SVC or the GP, some of this information are already given, for others this is hardly the case.*

The methods used in this work are indeed standard methods. In the initially submitted manuscript, we omitted more verbose explanations to keep the text concise. The reviewer noted an excellent point about the audience. We thus extended the explanation with advantages and disadvantages along with some references to the previous studies. Nevertheless, we tried to be as brief as possible. The updated manuscript is attached below (with equations omitted).

[revised manuscript text omitted]

*c) Especially for readers with a geoscientific background (as said, probably the majority of NHESS-readership) it would be interesting to read something about the relative importance of the various factors listed in Tab. 1. I understand that this may be quite different for the different classification schemes. But at least for those schemes eventually assessed to yield the best performance a qualitative summary could be listed,may mentioning the five most important factors in order of relevance.*

Based on this comment and the comment given by another Referee, we conducted a permutation feature importance analysis using the Gaussian processes (GP) model and the randomly selected test set of the national dataset. The same model and data are used to produce the case examples.

The manuscript is appended with the following chapters (page 17):

"The relevance of the individual predictive features can be explored by using the permutation test, as done by Breiman (2001). First, the baseline score of the fitted model is calculated using the test set. Then each feature is randomly permuted, and the difference in the scoring function is calculated. The random permutation is repeated 30 times for each parameter, and the average of the results is used. The procedure offers information on how important the feature, the individual parameter,

is to obtain good results. It should be mentioned that highly correlated features may get low importance as other features work as a proxy to the permuted feature. Using completely independent features is not, however, possible in weather data since weather parameters are often dependent on each other, and eliminating even the most apparent pairs from used features impaired the results in our experiments.

We used the macro average of F1 defined in Equation 7 as a scoring function and randomly selected test set from the national data. The relevance is shown in Figure 7. Most features show at least little relevance for the results. The first twelve features are more significant than the rest. The most important features contain at least one representative of all meteorological parameters used in training. In other words, all employed meteorological parameters are important for the prediction, while different aggregations are contributing to the "fine-tuning" of the model.

As Figure 7 shows, the most significant parameter regarding our model performance is the average wind speed. Numerous studies support our result of wind being the most important damaging factor (Virot et al., 2016; Valta et al., 2019; Jokinen et al., 2015) that are, however rather highlighting the importance of maximum wind gusts. Surprisingly, in our analysis, the wind gust speed does not belong to the most critical parameters. Instead, maximum mixed layer height, related to the wind gustiness, contributes crucially to the model performance. The dependencies between predictive features might be one reason for some parameters to have lower rank in the results.

The stand mean diameter and height are the most important features regarding the forest parameters, which corresponds to our expectations. Previous studies also state these features to influence the wind damage in forests (Pellikka and Järvenpää,2003) and hence indirectly electricity grids. As Pellikka and Järvenpää (2003) and Suvanto et al. (2016) discuss, also the age of the forest has an impact on storm damages. However, in the feature importance test, forest age does not seem to contribute significantly to the prediction outcome.

The most important object feature is the size of the object. Object movement speed and direction did not contribute to the results much. However, previous studies indicate that besides the size of the impacted area, the duration of strong winds – i.e., the movement speed of the system – influences also the amount of damage (Lamb and Knud, 1991)."

[Figure]

Figure7. Permutation feature importances using GP classification method trained with the randomly selected national dataset. The higher effect on the F1 score is (y-axis), the bigger is the significance.

*d) As far as I understand, the evaluation metrics in equations (4)-(7) are standard metrics used in the field of machine-learning based classification. However, what I am missing in these scores is any consideration of the distance between predicted and observed class. Clearly a prediction of "severe damage" in cases of no observed damage and vice versa is worse than predicting "small damage" in these cases. But this is not reflected in any penalty for the given scores. Maybe this is a wise solution given that the classes are very different in population. Otherwise an "algorithm" always predicting no damage might be superior with respect to a score taking this distance into account. I would ask the authors at least to comment on this matter and explain why they do not penalize larger distance between prediction and observation.*

This is an important notation. The used metrics are indeed selected to take the imbalance between classes into account. We also want to provide the results in well-known standard metrics to give the reader an intuitive image of the performance. The only metrics, which take the class distance into account, we are aware of, are Gandin and Murphy Equitable Score (GMSS) and its derivatives. However, these are relatively complicated metrics and not generally known. Thus, they would hence provide only a little value to the readers.

We commented on this matter in the updated manuscript following (on page 15, line 290-293):

"[...] The selected metrics do not take a distance between predicted and true class into account. It is naturally worse to predict, for example, class 0 (no damage) in the case of true class 2 (high damage) than in the case of true class 1 (low damage). We

decided, however, to use metrics that measure the method performance properly with imbalanced classes."

*e) I wonder why the authors decided to provide deterministic category predictions. This Is to some degree a philosophical discussion but given the nature of the task to make a prediction and further supported by i) the rather arbitrary distinction between event classes and ii) the large number of influential factors (some of them considered in the categorization schemes but many more existing in the real world), I wonder why the authors didn't design a scheme that provides probabilities for the distinct event categories. It is often argued that end-users prefer deterministic predictions but it is clearly a fact that predictions such as produced in this study are subject to significant uncertainty. So, why not making this uncertainty transparent by providing related estimates in the form of probabilities? I do not ask the authors to redesign their whole approach but please comment on this issue. Maybe it is worth considering this as a future extension for advancement of the presented approach.*

As well-argued, supplying the prediction with uncertainty information might be beneficial to some end-users if appropriately presented. Presenting the uncertainty in the correct way is also under broader discussion in meteorology due to wider use of the ensemble predictions. Providing uncertainty has, however, several challenges in this context:

1. As the referee already noted, at least the power grid operators prefer simple deterministic prediction. The simple view for the prediction is especially important in daily use where operators have only a little time to investigate the predictions.
2. The uncertainty would originate as a probabilistic prediction of the classification model, which describes the confidence of the model prediction instead of the reliability of the actual predictions. In other words, the uncertainty would not consider any sources of errors not introduced to the model. For example, the amount of leaves in the trees significantly affects the number of caused outages, but are not considered in the prediction due to shortcomings in available data. The model could predict an incorrect class with a very high confidence as it is not aware of tree leaves at all. Providing this kind of uncertainty would be easily misinterpreted by non-expert users. Similar effects can be seen in current ensemble prediction systems when the whole ensembles cluster is biased and true values are outside the confidence interval. Therefore, we argue that the performance metrics described in this work are better guidance for the prediction uncertainty.

Having said that, especially expert users like duty forecasters would benefit from the uncertainty information. We added the following future possibility to the updated manuscript (page 23, lines 441-444):

"End users, especially expert users like duty forecasters, would benefit from the uncertainty information originating as the probabilistic prediction of the classification

model. However, the presentation of such information should be very carefully chosen not to mislead non-expert users for overconfidence."

*f) A very general issue is that the authors use the term prediction (and so do I in this review) but in fact the presented approach is based on atmospheric REanalysisdata, i.e. it relies on data retroactively produced from observations. I would ask the authors to rephrase respective introductory and conclusive remarks in a way that it becomes clear that this study serves as a general introduction of this approach and a proof of concept while a quasi-operational implementation at weather services or power network providers would have to be based on actual weather predictions which will introduce additional uncertainty to the final product.*

> The term *prediction* is widely used in the field of machine learning in the meaning of model output. In this context, it may be confused with actual weather or outage prediction, which is not our meaning. We clarified this issue in the updated manuscript introduction following (page 3, lines 70-74):
>
> "[...] The ERA5 atmospheric reanalysis (European Centre for Medium-Range Weather Forecasts, 2017) provides the primary meteorological input data for this study, while the national forest inventory provided by The Natural Resources Institute Finland (Luke) is used to represent the forest conditions in the prediction. Finally, historically occurred power outages from two sources are used to train the model. However, the operational use of the model would require the use of weather prediction data instead of reanalysis."
>
> And also in conclusion following (page 22, line 412-414):
>
> "The evaluation was, however, based on the ERA5 reanalysis data. Using the method in operations would require weather prediction data, which introduces additional uncertainty to the outage prediction."

*g) Some of the figures need optimization. Please see my respective specific comments.*

> We went carefully through all figures and enlarged the labels.

*h) I am not a native English speaker myself, so I usually refrain from judging the language used in manuscripts written by others. However, in this example I have the strong impression that the language should be revised. A particular example are frequently missing definite and indefinite articles ("a" and "the"). Other examples can be found in my specific comments*

> We carefully checked the language and made corrections to the manuscript.

**Respond to the specific comments**

*1) line 14: Please revise your citation. This is certainly no person with the family name "Re" who is cited here but an institutional citation referring to a publication by the Munich Re.*

> We appreciate this note. We changed the citation and also added the URL address. (Page 1, line 14)

*2) lines 20-21: "...up to 69% compared to previous years". What is meant here? Is It an increase of 69% compared to previous years or a total of 69% of outages in 2011/2013v which are associated with windstorms. If the latter is the case, then please delete "compared to previous years".*

> When rereading this sentence, we acknowledge that it can be easily misunderstood. During the years of 2011 and 2013, the share of windstorm-induced outages was 69% of all outages. We deleted the last part of the sentence, as the referee suggested.

*3) line 27: "Ulbrich et al. (2009)", not "Ulbrich et al. (Ulbrich et al., 2009)"*

> We corrected this with compliments. (Page 2, line 25).

*4) lines 31-33: Please rephrase to make clear that this sentence contains references to studies contradicting the fore-mentioned studies and their results.*

> Pointing this out made us reread and clarify the entire paragraph. We reorganized it entirely in the following way. The update can be found in the updated manuscript on page 2, line 25-37:

> "As Ulbrich et al. (2009) describe, there is no scientific consensus on how the occurrence and magnitude of extratropical storms will evolve in the future. Based on existing literature the windstorm-related damages have increased and are increasing, while it remains unclear whether this is due to the increasing exposure of society or the number and intensity of extratropical storms. Gregow et al. (2017) discovered that windstorm damages have increased significantly during the past three decades, especially in northern, central, and western Europe. Also, several other studies suggest an increase in wind-related damages in Europe (Csillery et al. (2017), Haarsma et al. (2013), Gardiner et al. (2010)). Interestingly, some studies detected a decrease in the total number of extratropical storms (i.e. Donat et al. (2011)), while others found an increase in the number of extreme storms in specific regions, like western Europe and Northeast Atlantic (Pinto et al. (2013)). Another supporting view of a potential increase in extratropical storms in northern Europe can be found in the IPCC (2018) report. The report states extratropical storm tracks to being sifted towards the poles, which might affect the storminess in northern Europe. Thus, it may be concluded that also the losses related to extratropical storms are likely to increase especially in northern Europe. However, as Barredo (2010) emphasizes, the cause for increased losses can at least partly be explained by the increasing exposure of society rather than the increased number of windstorms."

*5) line 46: Delete "large-scale storms" and "small-scale storms" and just name the meteorological phenomena themselves as they are now listed in brackets. It is misleading to call hurricanes large-scale if then coming to the extra-tropical storms which are even larger in spatial scale.*

> *We changed this in the manuscript, as suggested.*

*6) lines 52-55: The purpose of this sentence is a little unclear to me, especially the reference to the IPCC-SREX-report. It's fine citing this report but not as one of several/many examples supporting this statement. It is basically the probably most comprehensive summary/review of studies indicating this.*

> This sentence and the reference is indeed detached from the previous sentences. We rephrased the end of the paragraph as follows (updated manuscript page 3, lines 58-61):
>
> "The framework of IPCC (2018) emphasizes that the impacts of extreme weather risks can be analyzed by estimating the hazard, vulnerability, and exposure. In an increasing manner, connecting these fields (i.e. the natural hazard with the societal factors) is done with machine learning (Chen et al. (2008))."

*7) line 64: Maybe replace "features" by "storm object features" or "storm object characteristics"*

> The features contain both storm and forest characteristics. We changed that into form (page 3, line 77):
>
> "[...] Chapter 3.2 considers storm and forest characteristics hereafter called features. [...]"

*8) lines 68-73: Please indicate the purpose of each dataset in this study, e.g. "the ERA5 atmospheric reanalysis (Hersbach et al., 2019) provides the primary meteorological input data for this study..."*

> Thank you. We clarified ERA5 and added additional sentences about other datasets and their roles. (Updated manuscript page 3, lines 70-74):
>
> "The ERA5 atmospheric reanalysis Hersbach et al., 2019, provides the primary meteorological input data for this study, while the national forest inventory provided by The Natural Resources Institute Finland (Luke) is used to represent the forest conditions in the prediction. Finally, historically occurred power outages from two sources are used to train the model. However, the operational use of the model would require the use of weather prediction data instead of reanalysis."

*9) lines 74-80: Please indicate explicitly which level you use regarding the ERA wind data. I guess it is the 10m-winds but this is not said here. Additionally, you may comment on the issue regarding ERA5 surface winds which is described at https://confluence.ecmwf.int/display/CKB/ERA5%3A+large+10m+winds . As far as Ican see this does not affect this study as all problematic occasions of unrealistic high wind speed happened at geographical locations far off the study domain. Still I consider this worth mentioning as some readers may not be aware of this issue in general (so the authors could contribute to a more widespread awareness of this problem) and others may be aware of the problem but not its location and related irrelevance for this study.*

The 10-meter wind gust from the surface data were used. We added this elaboration to the manuscript.

We added the following comment about the high wind speeds to the updated manuscript on page 4, line 93-95:

"ERA5 data are also known to contain unrealistically large surface wind speeds in some locations (European Centre for Medium-Range Weather Forecasts, 2019). None of these locations are, nevertheless, inside the geographical domain of this work."

*10) lines 84-88: What is the specific benefit of using the two local datasets on top of the national dataset for this study? Please comment.*

While containing basically the same information, they also differ significantly. The national dataset contains many more outages than the local datasets, but the outages are reported with lower geographical accuracy. We train our classification models with both datasets to evaluate their performance for different types of data.

We added this information to the updated manuscript on page 4, lines 109-115. We also improved Figure 1 and moved it in the data section based on another referee's comment.

"Figure 1 illustrates the geographical coverage of the power outage data. The local dataset contains all outages from 2010 to 2018 from the northern area (Loiste) and outages related to major storms in the southern area (JSE), shown in Figure 1a. The national dataset contains all outages in Finland from 2010 to 2018 divided into five regions, shown in Figure 1b. The national dataset contains in total 6 140 434 outages with relatively low geographical accuracy. On the other hand, the local dataset represents a substantially smaller geographical area with a good geographical accuracy but contains only 22 028 outages in total. We train our classification models, described in more detail in Chapter 3.4, with both datasets to evaluate their performance for different types of data."

[Figure]

**Figure 1.** (a) Geographical coverage of the outage data (local dataset). The red lines represent the power grid of Loiste (northern grid company) and the green lines the operative areas of JSE (southern grid company). Outages of the local dataset are collected from both of these areas. (b) Regions in the national outage dataset. Outages are gathered from the whole Finland and aggregated to the regions shown in the figure.

*11) lines 97-99: The reason given for using a threshold of 15m/s is valid as long as observed winds are considered. However, ERA5-winds are not observed winds, especially regarding gusts. It's basically model results. It should be noted here already, that at least a little bit of sensitivity tests have been performed yielding 15m/s to be the "best" choice. However, the motivation behind this study to develop a scheme which is applicable for quasi-operational forecasts would imply a transfer to a different source of meteorological data, basically weather predictions. Weather prediction models feature quite different distributions of surface wind speeds. Hence, for such an application a thorough test of the use of this threshold would be necessary. I would like to point out that there are approaches existing in the published literature on wind damage that make use of thresholds which are tied to the specific wind climatology of respective datasets, e.g. by making use of specific quantiles rather than absolute values.*

As the referee noted, the chosen threshold is supported by the previous studies (Gardiner 2013, Peltola 1999, Valta 2017) and empirical knowledge of the experienced duty forecasters and power grid operators.

We are aware that the optimal threshold depends on the chosen data source, and it is also highly dependent on the time of the year and other environmental conditions. As Peltola et al. (1999) discuss, even the specific tree species are sensitive and uprooted with different wind speed thresholds. During frozen ground and leafless periods, 15 m/s wind hardly harms any trees, but during summer months, when the trees have leaves, and the soil frost does not anchor the forest to the ground, 15 m/s can be already damaging. Thus, the used threshold depends on both data source and environmental factors, and is always a compromise.

As the referee suggests, using specific quantiles would be a proficient way to determine the correct thresholds. However, with an object-based approach, the use

of quantiles is not a straightforward task since the object needs to have the same absolute value inside the application domain to be a valid polygon. Therefore, the thresholds of the objects can not be always selected optimally.

We evaluated the method with a 20 m/s threshold with worse results. The evaluation is shortly mentioned in the initially submitted manuscript on page 13, line 246. However, trying out different thresholds between 15-20 m/s might yield better results. Unfortunately, this would be an intensive computing task requiring both time and budget.

We added a discussion about this matter to the updated manuscript on page 22, 415-430:

"The presented object-based approach has both advantages and disadvantages. Extracting storm objects in advance, preprocesses the data for machine-learning techniques, such as RFC, which do not perform feature learning. It enables machine-learning methods to focus only on the relevant parts of the data. Methods not containing feature learning, such as RFC and logistic regression, have been found to outperform neural networks for forest (Hart et al., 2019) and weather data (Tervo et al., 2019). It also leads to significantly faster training times. Processing objects instead of the grid makes it also easier to track and use object attributes such as age, speed, and movement. Moreover, objects are easy to visualize, and user interfaces may be enriched with related actions such as tracking and alarms.

On the other hand, storm objects use only aggregated attributes, which may decrease the classification accuracy when predictive features vary significantly under the storm object area. Several machine-learning methods, i.e., deep neural networks, could be trained to employ those local features to gain better accuracy. Such methods could also utilize three-dimensional data.

Extracting storm objects requires a fixed threshold of wind gust and pressure, which may vary depending on the characteristics of geospatial locations. Nevertheless, the previous studies indicate the critical threshold for wind gust speed to be the same for the almost whole geospatial domain of this work (Gardiner et al., 2013). Moreover, the correct threshold may vary depending on the data source. When extending the geospatial domain or changing the data source, this would become a more serious issue, and different thresholds might be needed. One possibility to determine the optimal threshold might be to use specific quantiles of the parameter values, but this would need further studies."

*12) line 103: Do you mean "...connected to objects in preceding timesteps"?*

Yes, this is what we mean. We updated the manuscript on page 5, line 135.

*13) line 103: Why do you call this "Algorithm 1" if there is no "Algorithm 2"? Why not simply calling it "Storm tracking algorithm"?*

We prefer this, possibly a little clumsy, naming to be consistent with figure and table naming and to give a clear hint for the reader about the reference to the separately described algorithm (shown on another page).

*14) line 103-104: Maybe I missed something but it seems to me you are not providing any information about the criterion to define/identify a "pressure object"*

Please see the answer to the next comment.

*15) line 103-104: You mention "the threshold" but such a threshold has not been introduced yet. This is done a few lines below. Please rephrase.*

We clarified the paragraphs describing the object identification and tracking method in the updated manuscript, page 5, lines 124-138:

"Storm objects are identified by finding contour lines of wind gust fields using 15 ms$^{-1}$ thresholds from the ERA5 surface level grid with a time step of 1 hour. The contouring algorithm is capable of finding interior rings of the polygons. The used wind gust fields did not, however, contain any such cases. Thus one storm object represents a solid area (polygon) where hourly maximum wind gust exceeds 15 ms$^{-1}$ during one particular hour. The threshold of 15 ms$^{-1}$ is selected as different sources indicate Finland being vulnerable to windstorms and rather moderate winds (from 15 ms$^{-1}$) causing damages to forests (Valta et al., 2019; Gardiner et al., 2013). Valta et al. (2019) developed a method to estimate the windstorm impacts on forests
by combining the recorded forest damages from the nine most intense storms and their observed maximum inland wind gusts. According to the formula developed in the study, the inland wind gusts of 15 ms$^{-1}$ alone result in forest damages of 1800 m$^3$.

We also identify pressure objects by finding contour lines using a 1000 hPa threshold to connect potentially distant wind objects around the low-pressure center to the same storm event.

After identification, storm objects are connected to other storm objects around the common low-pressure objects and to the storm and pressure objects in preceding timesteps using Algorithm 1. Each object having pressure objects or preceding objects within the threshold is assigned to the same storm event and gets the same storm ID. Single storm objects without nearby pressure objects or preceding objects are left without ID as they are not assumed to be part of any storm."

*16) line 111: "That means that wind objects are not assumed to move..."*

Please see the answer on the next comment.

*17) line 111: "45km" instead of "45km/h"; and please add "from one hourly timestep to another*

We appreciate these valuable and detailed suggestions and updated the sentence to form (page 6, line 141-143):

"[...] In other words, storm objects are not assumed to move over 200 km and pressure objects over 45 km from the preceding hourly time step (Govorushko, 2011)."

The term "wind object" was also changed to "storm object" based on the comment by another Referee to be consistent.

*18) line 115: "The first group is a number of object characteristics ... which are calculated ..." to the end of this sentence.*

Updated with compliments on page 6, lines 147-148.

*19) line 117-118: Please provide more details how you aggregate. Are the minimum/maximum/average values calculated over all grid boxes identified to belong to the storm object (i.e. exceeding 15m/s)*

Yes. We clarified this to the updated manuscript on page 6, line 149-151.

"To represent each parameter with one number, we aggregate values as a minimum, maximum, average, and standard deviation calculated over all grid cells under the object coverage."

*20) line 118: Replace "over" by "on"*

Replaced with thanks on page 6, line 151.

*21) line 119: Replace "features" by "characteristics"*

Replaced with thanks on page 6, line 152.

*22) line 120: Replace "in the damages" by "to the damages", "support" by "complement", and "with weather parameters" by "for weather parameters"*

Replaced with thanks on page 6, line 153.

*23) line 121: Replace "aggregated from" by "aggregated over".*

Replaced with thanks on page 6, line 154.

*24) line 124: Here you mention the samples for class 1 and 2 but the class definition has not yet been introduced. This happens in the next section. Please refer to this section and include a very brief definition of the two classes in this sentence, e.g."severe damage" and "small damage"*

> We restructured the text to introduce classes at the end of the Chapter on page 10, line 202 (originally on page 8, line 155).

*25) line 131: Now you introduce the general class definition (no damage, low damage,high damage) but again the exact definition is found at the very end of section 3.3. Additionally, the thresholds used to distinguish between the classes, especially between the two classes containing damage, seems to be completely arbitrary. AT least there is no reason given why the respective number of outages is considered to be low-damage or high damage.*

> We restructured the text to introduce classes on Chapter on page 10, line 202 (originally on page 8, line 155).

> The thresholds used in the class definitions are discussed more in response to comment 28.

*26) Fig. 1: Looking at the red lines in Fig. 1a & b I get the impression that only the lines for the northern local dataset illustrate actual power lines. The lines for the southern local dataset rather seem to be boundaries of sub-regions or so just as Fig. 1b contains region boundaries. I suggest to use different colors for different types of information. The spatial distribution of outages in Fig. 1c & d seems to having been smoothed. If so, please indicate this and the reason for doing so.*

> This is a valid point, and the other Referee pointed this out as well. The differences between the network topologies are simply explained by the data we have received from the two individual companies. From the northern company (Loiste), we received a shapefile of their grid. The southern company (JSE) provided their operational areas instead of the grid topology. Therefore, these two topologies look so different, even though JSE's grid also is similar compared to Loiste.

> We have now separated Figures 1a and 1b from 1c and 1d and improved the figures based on the suggestions of both referees. (Pages 5 and 9 in the updated manuscript).

[Figure]

**Figure 1.** (a) Geographical coverage of the outage data (local dataset). The red lines represent the power grid of Loiste (northern grid company) district and the green lines the operative areas of JSE (southern grid company). Outages of the local dataset are collected from both of these areas. (b) Regions in the national outage dataset. Outages are gathered from the whole Finland but aggregated to the regions shown in the figure.

The spatial distribution of the power outages has been produced as a spatial heatmap. In other words, it is represented as a density of outages. This visualization technique is selected to illustrate the spatial distribution of a large dataset as well as possible. We updated the figure based on the other referee's comment and clarified the visualization technique in the caption.

[Figure]

**Figure 2.** Spatial distribution of the outages between 2010 and 2018 visualised as a spatial heatmap. (a) JSE network (southern area) (b) Loiste network (northern area)

*27) lines 143-149 (and especially when reading lines 145-146): The reader immediately wonders why the authors stay with the 15m/s-threshold and why this is not analyzed in terms of quantitative measures. A simple example might be hit rates and false alarm rates or so. It is only in Sec. 4 (lines 248-250) that the authors write that storm identification with 15m/s yields a better basis for the following classification. Please Refer to this later explanation here.*

We referred to the explanation in the updated manuscript on page 10, line 191-192.

*28) lines 155-158: Eventually the class definitions seem to be set arbitrarily. If there is a reason behind the particular thresholds, please name these.*

We find that when designing new tools, especially impact forecasting/estimation tools, some arbitrary "first guesses" have to be taken. As mentioned in the manuscript, the limits are designed together with the power distribution companies and duty forecasters, and they aim to be as simple and intuitive as possible. However, power grid operators do not have any specific thresholds where the actions are taken. We are also not aware of any previous studies justifying any specific thresholds, especially in Finnish conditions.

The distinction between class 1 (low damage) and class 2 (high damage) is designed so that class 2 is truly exceptional. Class 2 represents roughly 20 percent of all samples, causing at least some damage and roughly 3 percent of all samples in both datasets.

Notably, the limits can be relatively easily changed in the future based on the end-users requirements or further research.

*29) lines 160-161: Why is centering and normalization necessary? Probably for some classification algorithms but not for all of them, right?*

The centering and normalization are necessary for all methods except the Random Forest Classification (RFC). RFC is a decision tree method which creates the splits based on the order of the values to each feature separately. Thus, the normalization and centering do not bother RFC either.

*30) lines 162-163: Please describe briefly what the application of SMOTE means and why this is beneficial/necessary.*

We added the following description about the SMOTE to the manuscript on page 12, lines 209-216:

"[...] To cope with the imbalanced class distribution, we generate artificial training samples using the synthetic minority oversampling technique SMOTE (Chawla et al., 2002). The SMOTE creates new training samples based on their k=5 nearest neighbors following:

$$x_{new} = x_i + \lambda \times (x_{zi} - x_i)$$

where $x_i$ is the original sample, $x_{zi}$ is one of $x_i$'s k-nearest neighbour and $\lambda$ is a random variable drawn uniformly from the interval [0,1]. After augmentation, all

classes have an equal number of samples, which reduces classification methods' tendency always to predict the majority class."

*31) lines 204-206: Why did you choose this specific topology? Did you test others? How is the sensitivity of the results to the networks topology?*

The topology was searched by iterating different combinations of topologies and hyperparameters and searching for the best possible results. We clarified this into the manuscript following (page 14, lines 270-276):

"We searched the correct model parameters and network topology for local and national datasets by running multiple iterations of random search 5-fold cross-validation to obtain the best possible micro average of F1-score (defined in Chapter 4) employing Talos library (Autonomio, 2020). The final setup composes of Nadam optimizer (Dozat, 2016), random normal initializer, and relu activation function for hidden layers. Binary cross-entropy was used as a loss function. Optimal network topology varied in different datasets. For the local dataset, the best results were obtained with a network containing three hidden layers with 75, 145, and 35 neurons. For the national dataset, the best results were obtained with a network containing three hidden layers with 75, 195, and 300 neurons. During the optimization process, the results varied between different setups from 0.6 to 0.95 in terms of F1-score."

As also stated in the updated manuscript, the results varied from 0.6 to 0.95 in terms of F1-score. KDE plot of the results from the final iteration of searching the best possible network topology for the local dataset is attached below as an example.

[Figure]

Figure: KDE plot of the results from the final iteration of hyperparameter and topology search.

*32) line 236: Please explain the content of the confusion matrices briefly. Again this is probably clear to people profound in machine-learning based classification but not necessarily to the general readership in geosciences. If I understand correctly, it is simply the ratio of cases for each observed class that is show in the cells for the respective predicted classes, right?*

We added the following information to the manuscript on page 19:

"Each cell of the confusion matrices represents a share of predictions having a corresponding combination of predicted and true class. For example, the middle right cell tells the share of samples belonging to class 1 but predicted to have class 2."

*33) Section 4.1: This whole section is where my major comment a) becomes visible. If I understand correctly, it is just the event as a whole which is assigned with the respective category, complemented by the polygon of the storm object(s). Is it possible that different objects of one specific event are assigned with different classes? Fig.6a seems as if this is possible. On the other hand the northeastern object is outside of Finland, so it is clear that there is no damage (to Finnish power lines) observed. In this context it becomes also visible that intra-object refinement of the classification would be desirable. It makes hardly any sense for a prediction of potential damage to power lines (due to windthrow in forests) that the storm objects extend over the Baltic Sea. I understand that this is due to the primary identification being solely based on the exceedance of the wind speed threshold. However, I ask the authors to thin and comment on my general comment a). Additionally, this case study validation refers to observed wind gusts when qualitatively assessing the credibility of these specific predictions. But the authors made it very clear that the potential damage due to windstorm depends on many more factors, partly non-meteorological but related to the forests themselves. This raises again the question of relative importance of the various factors.*

The classification is done to each storm object separately, and only power lines covered by the object are affected. Thus, the geographical areas can be distinguished in many cases. Furthermore, objects outside the area of coverage can be ignored.

Showing objects outside of Finland, for example, the Baltic Sea provides valuable information nevertheless to the operators in the form of preliminary information about approaching storms. The particular message in those cases is: The storm as it is now, would be (or would not be) hazardous to our power network if it was in our region. This gives the operator more tools and time to prepare.

We clarified the geographical coverage and the individual classification of storm objects in the introduction and method Chapter (please see the response to a general comment a).

We conducted a feature importance study and added it to the updated manuscript (response to general comment c).

*34) lines 306-307: This sentence ignores the fact that the actual study was based on reanalysis data. Using actual weather predictions - which would be necessary for this prospect mentioned here - would introduce additional uncertainty and very likely lead to worse results than derived in the current study. This does not lower the value of the current study but it is worth mentioning when writing about such potential quasi-operational applicability.*

> We added the following clarification to the updated manuscript on page 22, line 412-414:
>
> "The evaluation was, however, based on the ERA5 reanalysis data. Using the method in operations would require the use of weather prediction data, which introduces additional uncertainty to the outage prediction."

*35) line 309: Start the sentence with "Including data on..."*

> Modified with compliments.

*36) line 309: I agree that including data about forest soil and leaf index would probably be beneficial but it is questionable if such data is available in sufficient spatial and temporal resolution and coverage*

> The availability of such data is indeed questionable. We added a notation about this to the manuscript on page 22, lines 432-434:
>
> "Including data on soil moisture, soil temperature, and leaf index would most probably enhance the results, if available with sufficient spatial and temporal resolution since they would provide critical information about the environmental conditions."

*37) Appendix A: All text elements and axis labels in figures A1 and A2 are hardly readable.*

> We reduced the number of shown parameters to enlarge label size. We also replaced "speed_self", "angle_self", "area_m2", and "area diff" with corresponding feature names listed in Table 1. The updated figures are attached below:

[Figure]

Figure A1. Histogram of and fitted Gaussian distribution of selected predictive parameters in the local dataset. The Gaussian distribution is fitted separately to all samples and samples with little outages and many outages (classes 1 and 2 specified in Section 3.3).

[Figure]

Figure A2. Histogram of and fitted Gaussian distribution of selected predictive parameters in the national dataset. The Gaussian distribution is fitted separately to all samples and samples with little outages and many outages (classes 1 and 2 specified in Section 3.3).

---

## Author Comment (AC3) · 8 Oct 2020

**Response to the Comments SC1 on Manuscript Predicting power outages caused by extratropical storms**

Corresponding author: Roope Tervo, roope.tervo@fmi.fi
Oct 7th, 2020

We are thankful and flattered for the commentator for reading our paper carefully and providing us valuable improvement suggestions. In the following replies, we have addressed the comments and made the improvements in the manuscript.

The commentators input is indented and with italic typesetting. The authors' comments are with normal typesetting. Direct quotes from the manuscripts are marked with double-quotes.

**Responds to the general remarks**

*A thoroughly interesting paper. The methodology for identifying storms is especially interesting. However, there may be a few ways to improve the work presented. More specifically:*

*1) In lines 46 to 48, the authors claim that modeling power outages caused by extratropical events is an understudied problem. However there are actually several papers that describe a power outage prediction system designed specifically for modeling power outages from extratropical storms that are not cited: Yang et al, https://www.mdpi.com/2071-1050/12/4/1525; and Cerrai et al, https://ieeexplore.ieee.org/abstract/document/8656482*

> We appreciate this advice and added mentioned papers to the previous work. (Updated manuscript page 2, lines 52-54).

*2) In figure 4b, it's unclear why the data contains prominent examples where there are very few or no outages, but have a large number of customers affected. Is this trend real, or is it an artifact of noise in the data?*

> Only six customers usually lose electricity in one outage in the national datasets. In some cases, however, outages affect many more customers. We can not ensure the correctness of each data point, but we did check some extreme cases. Typically these cases occur in urban areas and are rare because the power network is mainly underground in these areas.

> We added a comment about this matter to the updated manuscript in page 10, lines 198-199:
> "Notably, in some rare cases, many more customers are affected. Based on our random inspections, these cases occur typically in urban areas and are rare because the power network is mainly underground in these areas."

*3) By using week as a predictor variable the authors may be over-fitting. For example, to my knowledge, there's no specific mechanism of why a storm on the 42nd week of the year would be particularly strong. But if you had several examples of strong storms on that week, the model would learn that trend and begin to predict strong outages just because of the week, independent of the actual meteorological characteristics of the storms. There are probably other, less problematic ways to describe seasonal aspects of storms to the model.*

This is truly a very valid concern. During the review process, we conducted a permutation feature importance analysis using the Gaussian processes (GP) model and randomly selected test set of the national dataset. The results of the analysis are shown below. Please consult responses to RC1 or RC2 for more information.

[Figure]

The results indicate that permuting week during the training process had only a little effect (0.015 +/- 0.001) on F1 score. Moreover, we conducted the feature importance study also using corresponding train data. In that case the week had almost the same effect (0.013 +/- 0.002) on the F1 score, which also indicates that using the week as a predictor does not lead to overfitting.

*4) I would recommend a more rigorous and comprehensive method for validating the model. As discussed in the paper, the k-fold cross-validation approach may not sufficiently isolate temporally or spatially correlated information from the model, and thus inflate the model's performance. The 2010 to 2011 holdout approach is presented as an alternative to this approach, but the types of storm events that occur often vary widely from year to year. A leave-one-day/week/month/year-out cross validation (where foreach day, week, month, or year in the database you hold out that data, train the model on the remaining data, and predict on the withheld data. Then evaluate the model on all of those results) would provide more compelling results.*

Thank you for the comment.

First, we would like to clarify that the test results does not represent k-fold cross-validation but randomly selected holdout as stated in the beginning of Chapter 4 (page 12, lines 208-209 in the originally submitted manuscript).

The validation could still be extended to more rigorous methods like the one suggested by the reviewer. We are aware of a potential autocorrelation issue when selecting the testset randomly. We selected to address this issue by solid year holdout since based on the data analysis (for example Figure 2, attached also below) 2010 to 2011 represents the whole data relatively well in terms of number of outages and storm objects.

We would also like to note that there is no significant difference between two different testset (randomly picked and continuous holdout). Thus, we believe that our method provided sufficient validation scores.

[Figure]

**Figure 2.** Storm object time series (15, 20 and 25 m s$^{-1}$ contours) with occurred outages for local and national datasets.

Nevertheless, we commented the issue in the Discussion section (page 22, lines 436-438 in the updated manuscript) :

"Especially in the randomly selected test set, data may be autocorrelated, which may lead to unrealistically good results. We have addressed this issue by also using continuous holdout from 2010 to 2011 for the test set. The evaluation could also be extended by, for example, a leave-one-day-out or leave-one-week-out method where for each week one day or for each month one week is hold out for validation purposes."

---

## Author Response (AR2)

**Response to the second comments on Manuscript Predicting power outages caused by extratropical storms**

Corresponding author: Roope Tervo, roope.tervo@fmi.fi
Oct 11th, 2020

We thank the reviewers for reading our paper and author's reply carefully and are thankful for the precious and detailed comments. In the following replies, we address the second comments and specify the improvements in the manuscript.

The referee's comments are indented and with italic typesetting. The authors' comments are with normal typesetting. Direct quotes from the manuscripts are marked with double-quotes.

**Responds to the general remarks**

*This is my 2nd round review of the manuscript "Predicting power outages caused by extratropical storms" by Tervo et al. I highly appreciate the comprehensive revisions performed by the authors. I think that they handled all of the reviewer comments as well as the short comments issued during the interactive discussion appropriately. The manuscript improved in all aspects that were critized in the first review. I specifically appreciate that the authors now provide a little more background information regarding the different classification algorithms and that they included a sensitivity analysis, illustrating the relevance of individual parameters for the best performing algorithm. I hence recommend publication of this manuscript. However, I have a few comments regarding some answers they made to my previous issues. These don't necessarily need to be taken up by the authors in terms of revising the manuscript but I would like to ask the authors to think and respond on these points.*

*1. When answering to my previous general comment (e), the authors state:*
*"[...] The uncertainty would originate as a probabilistic prediction of the classification model, which describes the confidence of the model prediction instead of the reliability of the actual predictions. In other words, the uncertainty would not consider any sources of errors not introduced to the model. For example, the amount of leaves in the trees significantly affects the number of caused outages, but are not considered in the prediction due to shortcomings in available data. The model could predict an incorrect class with a very high confidence as it is not aware of tree leaves at all. [...]".*
*All of the above is correct but the conclusion of the authors is wrong. In fact it is state-of-the-art and common practice in weather and climate prediction to check if a forecast system's uncertainty is in line with the forecast error. And when it is about probabilistic (ensemble based) forecasts this applies also to the reliability of the probabilistic prediction. In case of a "perfectly reliable" prediction system, the predicted probability matches the average (climatological) occurence of the specific event for all cases when the event was forecasted. And there are proper verification scores that can test for these characteristics. So there are ways to check that beforehand and probabilistic prediction systems that are not perfectly reliable in that sense can usually be corrected via calibration.*

*As written before: I do not expect that the authors revise their whole approach. I am fine with the comment they added to the manuscript in this respect. I just wanted to clarify this here.*

This is excellent clarification. We appreciate this.

*2. When answering to my previous specific comment (11), the authors state:*
*"[...] As the referee suggests, using specific quantiles would be a proficient way to determine the correct thresholds. However, with an object-based approach, the use of quantiles is not a straightforward task since the object needs to have the same absolute value inside the application domain to be a valid polygon. Therefore, the thresholds of the objects can not be always selected optimally. [...]"*
*This is a rather cheap excuse. It would be no problem to advance the identification of the storm objects in a sense that not a fixed threshold is used for all grid-boxes but instead a 2D-field containing individual thresholds for the respective grid boxes, still assigning simple boolean values to the resulting object field. In theory even 3D would be possible to account for the seasonality that was mentioned by the authors themselves.*
*I would like to make the authors aware of the fact that such an algorithm for identifying and tracking of storm fields based on a 2D-quantile-field already exists: The approach that was first introduced by Leckebusch et al. (2008) and since then used in multiple studies on extra-tropical storms (and recently even for tropical storms) is based on exceedances of the local climatological 98th percentile of surface-near wind speed. A detailed description of the complete algorithm is available only from my own PhD-thesis (Kruschke, 2015) but this general principle is named in all papers (in the order of 20-30 or so) based on this algorithm. I am not fishing for any citations here. and I would name other algorithms containing such a feature if I knew some doing so. It is fine with me that the authors proceed with their algorithm as it is, given that they included a hint towards quantiles being a possibility for optimization. I just wanted to make clear that such a modification is not impossible as indicated by the authors in their response.*

Our apologies for slightly misunderstanding the original comment. Forming storm objects based on the precalculated storm severity index would indeed be a prominent approach to adapt the threshold. The overall effects of such an approach in this particular application would, however, need further investigation.

We modified the manuscript following (page 22, lines 418-422):

"The fixed threshold of wind gust and pressure were used to extract the storm objects in this paper. Although the previous studies indicate the critical threshold of wind gust speed to be the same for the almost entire geospatial domain of this work (Gardiner et al., 2013), it would be beneficial to adapt the threshold based on the geographic location using, for example, storm  severity index (SSI) originally introduced in Leckebusch et al. (2008). Moreover, the correct threshold may vary depending on the data source."

*3. Closely related to the above, when the authors reply to my specific comment (33), they state:*
*" [...] Showing objects outside of Finland, for example, the Baltic Sea provides valuable information nevertheless to the operators in the form of preliminary information about approaching storms. The particular message in those cases is: The storm as it is now, would be (or would not be) hazardous to our power network if it was in our region. This gives the operator more tools and time to prepare. [...] "*
*In principle I agree but I expect that using a fixed threshold results into identifying comparably many storm objects over the Baltic Sea that disappear, once they reach land areas when surface winds are decelerated due to the higher surface roughness, orography and so on. In that case the statement of the authors that such a storm would cause damage over land is not really accurate and (assuming that my expectation is right) frequent warnings regarding storms over the Baltic without any further classification of severity may even lead to ignorance of such cases by power network operators. I don't see how the current algorithm is able to distinguish between "really" dangerous storms approaching over the Baltic Sea and those that can be expected to yield wind speeds over 15 m/s only over the sea, not over land.*

> It is true that many storm objects, visible over Baltic, vanish when they reach the continent.
>
> Nevertheless, the proposed method is not completely incapable to distinguish really hazardous storms from others. It may exploit object size, movement, and weather parameters to determine the damage potential of the storm.